# Plasmodium falciparum has evolved multiple mechanisms to hijack human immunoglobulin M

Chenggong Ji[1,2,6], Hao Shen [1,6], Chen Su[1], Yaxin Li[1], Shihua Chen[3], Thomas H. Sharp [4] & Junyu Xiao [1,2,5] ✉

Plasmodium falciparum causes the most severe malaria in humans. Immunoglobulin M (IgM) serves as the first line of humoral defense against infection and potently activates the complement pathway to facilitate P. falciparum clearance. A number of P. falciparum proteins bind IgM, leading to immune evasion and severe disease. However, the underlying molecular mechanisms remain unknown. Here, using high-resolution cryo-electron microscopy, we delineate how P. falciparum proteins VAR2CSA, TM284VAR1, DBLMSP, and DBLMSP2 target IgM. Each protein binds IgM in a different manner, and together they present a variety of Duffy-binding-like domain-IgM interaction modes. We further show that these proteins interfere directly with IgM-mediated complement activation in vitro, with VAR2CSA exhibiting the most potent inhibitory effect. These results underscore the importance of IgM for human adaptation of P. falciparum and provide critical insights into its immune evasion mechanism.

Malaria is one of the greatest killers of humankind throughout history and remains a major public health problem: approximately 241 million malaria cases were documented worldwide in 2020, resulting in 627,000 deaths[1]. Malaria is caused by infection with Plasmodium parasites, among which Plasmodium falciparum causes the most devastating disease. The merozoite form of P. falciparum invades the red blood cells to replicate inside, and the infected red blood cells (iRBCs) are eventually ruptured to release more merozoites, resulting in fever and hemolytic anemia. Furthermore, iRBCs can adhere to the placenta and brain endothelium, leading to fatal complications known as placental and cerebral malaria.

Immunoglobulins are central components of the immune system and provide critical protections against various pathogens, including P. falciparum. The immunoglobulin M (IgM) type of antibodies is the first to be produced in a humoral immune response[2,3]. The predominant form of IgM is an asymmetrical pentamer, with five IgM monomers joined together by the joining chain (J-chain)[4–6]. The presence of ten antigen-binding sites within an IgM pentamer allows it to bind and neutralize pathogens effectively. Furthermore, IgM efficiently activates the complement pathway, which plays a crucial role in malaria immunity[7].

During the evolutionary arms race between the Plasmodium parasite and humankind, P. falciparum has evolved strategies to antagonize the function of IgM. Plasmodium falciparum erythrocyte membrane protein 1 (PfEMP1) is a family of ~60 virulent proteins secreted by P. falciparum to the iRBC surface. PfEMP1 proteins have very large extracellular segments, consisting of different numbers and types of Duffy-binding-like (DBL) domains and cysteine-rich interdomain regions. These versatile modules endow PfEMP1 proteins with the ability to interact with a range of molecules in humans[8–10]. For example, VAR2CSA, a major culprit in placental malaria, can bind to chondroitin sulfate A (CSA) glycosaminoglycans, resulting in the sequestration of iRBCs within the placenta[11].

[1]State Key Laboratory of Protein and Plant Gene Research, School of Life Sciences, Peking University, Beijing, China. [2]Changping Laboratory, Beijing, PR China. [3]Joint Graduate Program of Peking-Tsinghua-NIBS, Academy for Advanced Interdisciplinary Studies, Peking University, Beijing, China. [4]Department of Cell and Chemical Biology, Section Electron Microscopy, Leiden University Medical Center, 2300 RC Leiden, The Netherlands. [5]Peking-Tsinghua Center for Life Sciences, Peking University, Beijing, China. [6]These authors contributed equally: Chenggong Ji, Hao Shen. ✉e-mail: junyuxiao@pku.edu.cn

TM284VAR1 is a PfEMP1 protein isolated from a cerebral parasite strain[12]. Like VAR2CSA, TM284VAR1 can cause rosetting, namely, the adhesion of iRBCs to uninfected RBCs. It is highly likely that TM284VAR1 contributes significantly to the virulence of this cerebral malaria strain. Importantly, both VAR2CSA and TM284VAR1 can interact with IgM; and it has been demonstrated that VAR2CSA employs IgM as a shield to conceal itself from immunoglobulin G (IgG) antibodies[13,14]. Similarly, a number of other PfEMP1 variants bind to IgM[15–17], and the presence of nonimmune IgM on iRBCs correlates with severe malaria[18]. In addition, DBLMSP and DBLMSP2, two *P. falciparum* proteins that do not belong to the PfEMP1 family, are also capable of interacting with IgM[19]. Both of these proteins comprise a single DBL domain that is responsible for binding to IgM and a SPAM (secreted polymorphic antigen associated with merozoites) domain that is involved in oligomerization[20]. In contrast to the PfEMP1 proteins that reside on iRBCs, these two proteins are located on the surface of *P. falciparum* merozoites[21]. It is likely that they also recruit IgM to provide camouflage for merozoites and thereby facilitate their evasion of IgG antibodies[19].

In this work, we present the cryo-electron microscopy (cryo-EM) structures of VAR2CSA, TM284VAR1, DBLMSP, and DBLMSP2 complexed with human IgM core. Our results uncover diverse modes of IgM targeting by these proteins, and shed light on immune evasion of *P. falciparum* facilitated by IgM.

# Results

## *P. falciparum* proteins bind to human IgM core

To understand how these *P. falciparum* proteins specifically bind IgM, we prepared the ectodomains of VAR2CSA (from the FCR3 strain) and TM284VAR1, as well as the DBL domains of DBLMSP (from field isolate 017) and DBLMSP2 (from the 3D7 strain) (Fig. 1a), and tested their interactions with the human pentameric IgM core (Fcμ–J) that consists of the IgM-Fc (Fcμ) pentamer and the J-chain[4]. Surface plasmon resonance (SPR) analyses demonstrate that each recombinant protein binds to Fcμ–J with high affinity, exhibiting $K_d$ values of 7–30 nM (Fig. 1b). These results confirm previous findings of IgM-recruiting abilities of these proteins, and demonstrate that their interactions with IgM do not require the presence of the antigen-binding fragments of IgM.

## Cryo-EM structure determination

We subsequently reconstituted the complexes between these *P. falciparum* proteins and Fcμ–J (Supplementary Fig. 1) and determined their cryo-EM structures (Fig. 2, Supplementary Figs. 2–5, Supplementary Table 1). Although some PfEMP1 proteins can bind IgM in a 2:1 ratio[22,23], 1:1 complexes were most clearly resolved for the four *P. falciparum* proteins investigated in this study. In all these structures, Fcμ–J exhibits a pentameric architecture, with the J-chain conferring asymmetry on the central Fcμ platform, as seen in the complex with the secretory

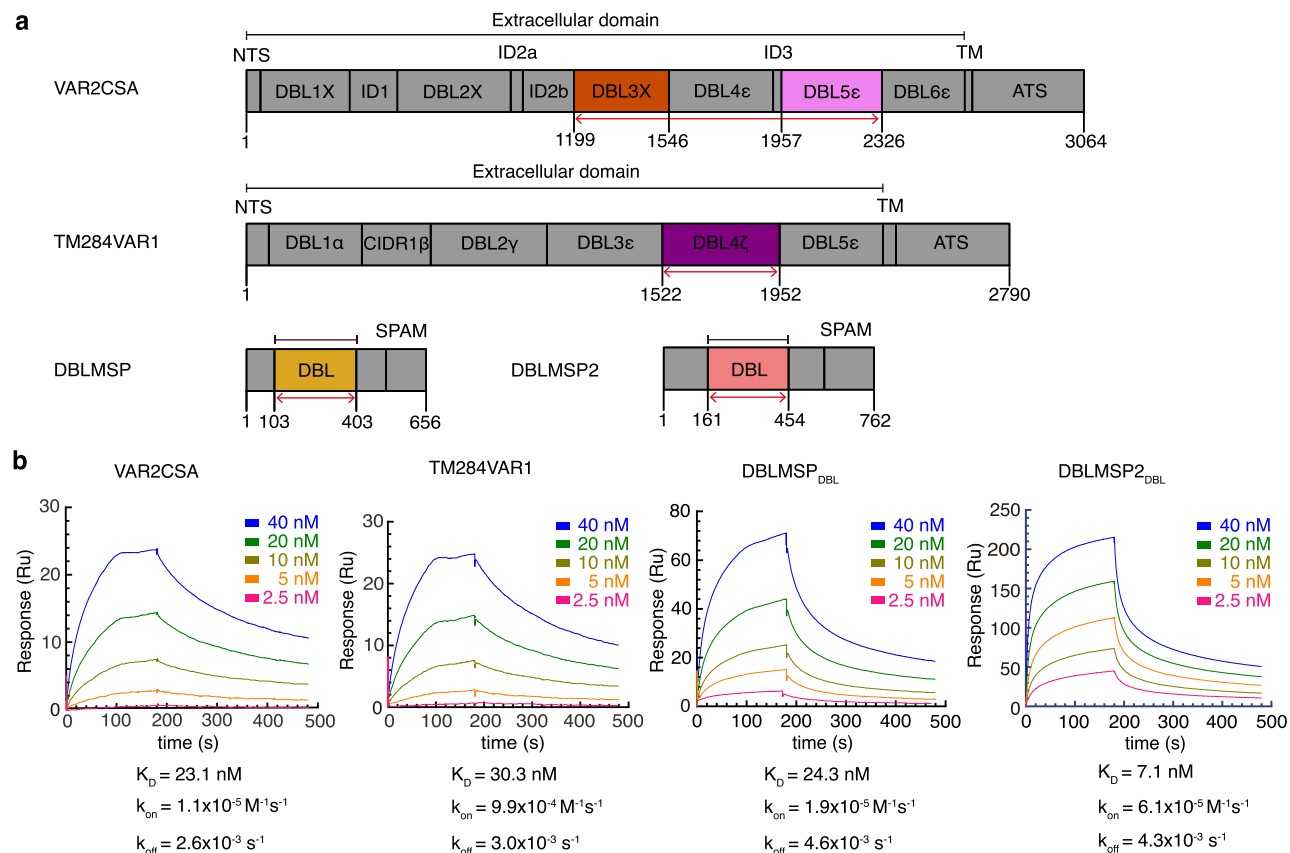

**Fig. 1 | The *P. falciparum* proteins directly interact with the human pentameric IgM core. a** Schematics of the domain organizations of VAR2CSA, TM284VAR1, DBLMSP, and DBLMSP2. Black lines indicate the protein fragments that are recombinantly produced for cryo-EM study, whereas red arrows indicate the regions that are structurally modeled into the density maps. Domain numbers are used to indicate DBL subclasses rather than positions in the gene in the recent nomenclature system[55,56]. For example, VAR2CSA domains are now referred to as DBLpam1–DBLpam2–CIDRpam–DBLpam3–DBLεpam4–DBLεpam5–DBLε10, whereas TM284VAR1 domains are DBLα1.8–CIDRβ2–DBLγ7–DBLε11–DBLζ2–

DBLε6. Nevertheless, old naming schemes are still adopted in this paper, since they were widely used in the previous literature. NTS N-terminal sequence, ID interdomain, TM transmembrane, ATS acidic terminal sequence, CIDR cysteine-rich interdomain region. **b** SPR analyses of the interactions between the *P. falciparum* proteins and Fcμ–J, performed by passing purified Fcμ–J (from 40 nM to 2.5 nM indicated with different colors) to immobilized *P. falciparum* proteins. All SPR experiments in this paper have been repeated at least two times with similar results.

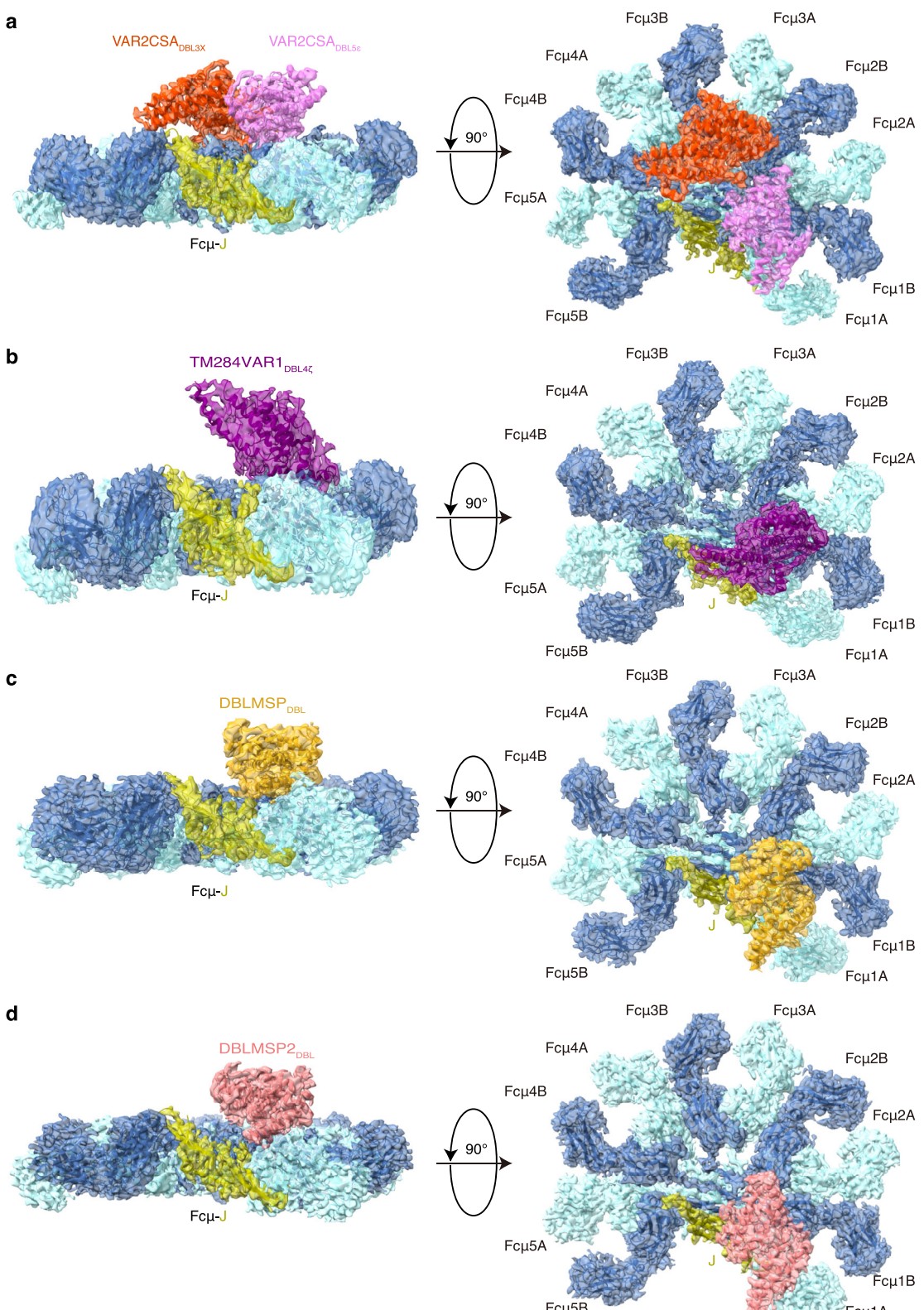

**Fig. 2 | Cryo-EM structures of the *P. falciparum* proteins bound to the human pentameric IgM core. a** Structure of the VAR2CSA–Fcμ–J complex shown in two orientations. For clarity, only the DBL3X (orange red) and DBL5ε (purple) domains of VAR2CSA that are directly involved in binding to Fcμ–J are shown. The Fcμ molecules are shown in two shades of blue, whereas the J-chain is shown in yellow.

**b** Structure of the TM284VAR1$_{DBL4\zeta}$–Fcμ–J complex. The DBL4ζ domain is shown in dark purple. **c** Structure of the DBLMSP$_{DBL}$–Fcμ–J complex. The DBL domain is shown in gold yellow. **d** Structure of the DBLMSP2$_{DBL}$–Fcμ–J complex. The DBL domain is shown in pink.

component (SC), i.e., the ectodomain of the polymeric immunoglobulin receptor (pIgR)[4,5]. The pIgR/SC-binding face of Fcμ–J is also targeted by the *P. falciparum* proteins, and the interactions between the *P. falciparum* proteins and Fcμ–J exclusively involve the Fcμ·Cμ4 domains, which is consistent with previous analyses[12,13,19]. The DBL domains in the *P. falciparum* proteins are responsible for interacting with Fcμ; interestingly however, they display distinct Fcμ-binding modes.

## Structure of the VAR2CSA–Fcμ–J complex

The 300 kDa ectodomain of VAR2CSA consists of six DBL domains plus the interdomain regions (IDs) (Fig. 1a). Recent cryo-EM studies demonstrated that the regions encompassing DBL2X–ID3 assemble into a stable core, whereas DBL5ε–DBL6ε forms a flexible arm[24–26]. Densities are present for the majority of this large molecule in the VAR2CSA–Fcμ–J complex (Supplementary Fig. 2a). Compared to the VAR2CSA structure determined in the absence of IgM, a large swing of the DBL5ε–DBL6ε arm can be observed (Supplementary Fig. 2f). The IgM-binding sites in VAR2CSA have previously been variously mapped to DBL2X, DBL5ε, and DBL6ε[27,28]; however, our structure unambiguously reveals that DBL3X and DBL5ε conjointly mediate binding to the Fcμ platform (Fig. 2a). This result is highly concordant with previous observations showing that IgM specifically excludes the binding of DBL3X- or DBL5ε-specific IgGs to RBCs infected by VAR2CSA-expressing *P. falciparum* parasites[13,29]. The major CSA-binding pocket is formed by several domains within the stable core of VAR2CSA, especially DBL2X and DBL4ε. DBL3X and DBL5ε are located distal to the CSA-binding site; therefore VAR2CSA should be able to bind to IgM and CSA simultaneously (Supplementary Fig. 2f, Fig. 3a). Indeed, previous studies showed that IgM did not affect the adhesion of VAR2CSA-bearing iRBCs to CSA[13].

DBL3X and DBL5ε together interact with three Fcμ units within the Fcμ pentamer (Fig. 2a). Both DBL3X and DBL5ε exhibit an archetypical DBL fold that can be further divided into three subdomains[30,31]: SD1 comprises mostly loops, whereas SD2 and SD3 contain a characteristic four-helix bundle and double-helix hairpin, respectively (Fig. 4a, b; Supplementary Fig. 6). DBL3X interacts with the Cμ4 domains of Fcμ2B (the ten Fcμ chains in the Fcμ pentamer are named as previously described, starting from the Fcμ chain that interacts with the C-terminal hairpin of the J-chain as 1A[4]) and Fcμ3B using residues in subdomain SD1 (Fig. 3b). Tyr1282 is sandwiched between Fcμ2B and Fcμ3B and packs against $Arg491_{Fcμ2B}$ and $Asn465_{Fcμ3B}$. $Arg491_{Fcμ2B}$ is also contacted by Asp1279 and Ser1281. Gln1231 and Thr1234 appear to contact $Ser524_{Fcμ3B}$ and $Glu526_{Fcμ3B}$, respectively; whereas Lys1238 forms an ion pair with $Glu532_{Fcμ3B}$. DBL5ε, on the other hand, interacts with Fcμ1B and Fcμ2B, also mainly using a loop in SD1 (Fig. 3c). Arg2050 packs with $Asn529_{Fcμ1B}$–$Thr530_{Fcμ1B}$, and also coordinates $Glu526_{Fcμ1B}$. Pro2055 and Ala2056 insert between Fcμ1B and Fcμ2B, and pack with $Arg491_{Fcμ1B}$ and $Asn465_{Fcμ2B}$–$Leu466_{Fcμ2B}$, respectively. Asn2057 forms a hydrogen bond with $Leu466_{Fcμ2B}$. Arg2059 interacts with $Glu525_{Fcμ2B}$ and $Asn529_{Fcμ2B}$. A VAR2CSA heptamutant (VAR2CSA-M), K1238A/D1279A/Y1282A/R2050A/P2055G/N2057A/R2059A, failed to interact with Fcμ–J, validating the functional relevance of the molecular interactions described above (Fig. 3d). Notably, VAR2CSA-M was purified well and eluted as a monodisperse peak on size-exclusion chromatography (Supplementary Fig. 1f), suggesting that the overall structure of this mutant is not disrupted.

IgM is a potent activator of the classical complement pathway; however, it has long been documented that the recruitment of IgM onto iRBCs by VAR2CSA does not render iRBCs susceptible to complement-dependent cytotoxicity[13]. In fact, another PfEMP1 protein, IT4VAR60, binds to IgM and blocks the deposition of C1q (a key component of the complement C1 complex) on the iRBCs, thereby protecting the iRBCs from complement-mediated lysis[23]. To examine whether VAR2CSA directly inhibits complement-dependent

cytotoxicity, we prepared a recombinant anti-CD20 IgM molecule by engineering the antigen-binding fragment of rituximab, a monoclonal antibody against CD20, onto Fcμ (Supplementary Fig. 1k). Indeed, this IgM molecule robustly triggered the lysis of CD20⁺ OCI-Ly10 cells in the presence of human serum complement (Supplementary Fig. 1l). In contrast, preincubation of this IgM with the ectodomain of VAR2CSA greatly reduced its ability to activate complement-dependent cytotoxicity, with a half maximal inhibitory concentration ($IC_{50}$) of 1.9 nM (Fig. 3e). VAR2CSA-M displayed no such inhibitory effect, further corroborating our structural and biochemical analyses.

## TM284VAR1$_{DBL4ζ}$ is responsible for interacting with Fcμ–J

The ectodomain of TM284VAR1 exhibits a flexible structure (Fig. 5a). A 3D reconstruction at 18 Å reveals an elongated architecture that resembles VAR2CSA to some extent (Fig. 5b). Although the entire ectodomain was used to reconstitute the complex with Fcμ–J (Supplementary Fig. 1b), only the DBL4ζ domain can be clearly visualized in the density map (Supplementary Fig. 3). Other regions likely display conformational disorder and are thus not discernible after single-particle averaging. This is highly concordant with previous biochemical analyses demonstrating that the DBL4ζ domain in TM284VAR1 is solely responsible for binding to IgM[12,32].

TM284VAR1$_{DBL4ζ}$ interacts with Fcμ1B, Fcμ2A, and Fcμ2B using residues from both SD1 and SD2 (Figs. 2b, 4c, Supplementary Fig. 6). In particular, the α3 helix and the following loop within the SD2 four-helix bundle provide a focal point for the TM284VAR1$_{DBL4ζ}$–Fcμ interaction (Fig. 5c). Because of this tight interaction, this region displays high-quality density with clear side chain features (Supplementary Fig. 3h). Glu1705 binds to $Arg491_{Fcμ1B}$. Arg1706 forms a bidentate interaction with $Glu468_{Fcμ2B}$. Lys1709 and Arg1712 interact with $Asp453_{Fcμ2A}$. Asp1716 and Asn1717 interact with the main chain groups of $Ala542_{Fcμ2A}$ and $Leu449_{Fcμ2A}$. Two TM284VAR1 mutants, E1705A/R1706A/K1709A (M1) and E1705A/R1706A/D1716A (M2), were generated to confirm the critical functions of these residues in binding to IgM. Both mutants fold well (Supplementary Fig. 1g, h), but display diminished interactions with Fcμ–J (Fig. 5d). Similar to VAR2CSA, the ectodomain of TM284VAR1 suppressed IgM-mediated complement activation, although not as potently, displaying an $IC_{50}$ of 52.9 nM (Fig. 3e). In contrast, neither TM284VAR1-M1 nor TM284VAR1$_{DBL4ζ}$ exerted such an effect.

The Tyr1728–Tyr1732 loop in TM284VAR1$_{DBL4ζ}$ also packs intimately with the Gly492–Pro497 loop in Fcμ1B (Fig. 5e). Indeed, bovine and mouse IgM differ significantly from human IgM at the Gly492–Pro497 loop, and neither of them binds to TM284VAR1[12,28]. Additionally, the replacement of the human IgM residues Pro494–Pro497 with the corresponding mouse sequence also abolished the interactions with TM284VAR1$_{DBL4ζ}$[12].

## DBLMSP and DBLMSP2 bind Fcμ using all three subdomains

Full-length DBLMSP protein is unstable and tends to form heterogeneous oligomers in solution (Supplementary Fig. 1c). Interestingly, full-length DBLMSP, but not the monomeric DBLMSP$_{DBL}$ domain, interferes with IgM-mediated complement activation (Fig. 3e). Therefore, the SPAM domain that is responsible for oligomer formation[20] is needed for this activity. The apparent $IC_{50}$ value (~119 nM) could be an underestimate due to the unstable nature of full-length DBLMSP. We further performed the cryo-EM study using DBLMSP$_{DBL}$ and determined the complex structure with Fcμ–J (Supplementary Fig. 4). Different from the DBL domains in VAR2CSA and TM284VAR1, DBLMSP$_{DBL}$ interacts intimately with Fcμ1A, Fcμ1B, and Fcμ2B using all three subdomains (Figs. 2c, 4d, Supplementary Fig. 6). In SD1, a loop involving DBLMSP residues Ile140–Ala143 inserts into the groove between $Cμ4_{Fcμ1A}$ and $Cμ4_{Fcμ1B}$, whereas His173–Arg174 contact $Cμ4_{Fcμ2B}$ (Fig. 6a). In SD2, Asp229–Ile232 pack with the 525–530 helix

in Cμ4_{Fcμ1B} (Fig. 6b). Glu352, Asn356, and Arg357 in SD3 engage Cμ4_{Fcμ1A} (Fig. 6c). Indeed, two DBLMSP_{DBL} mutants, N169A/H173A/R174A (M1) and D229A/Y230A/Q231A (M2), do not bind to Fcμ–J (Fig. 6d).

We also determined the cryo-EM structure of the DBLMSP2_{DBL}–Fcμ–J complex (Supplementary Fig. 5). When compared to the DBLMSP_{DBL}–Fcμ–J structure, an overall similar pattern of interaction between DBLMSP2_{DBL} and Fcμ–J is observed (Fig. 2c, d).

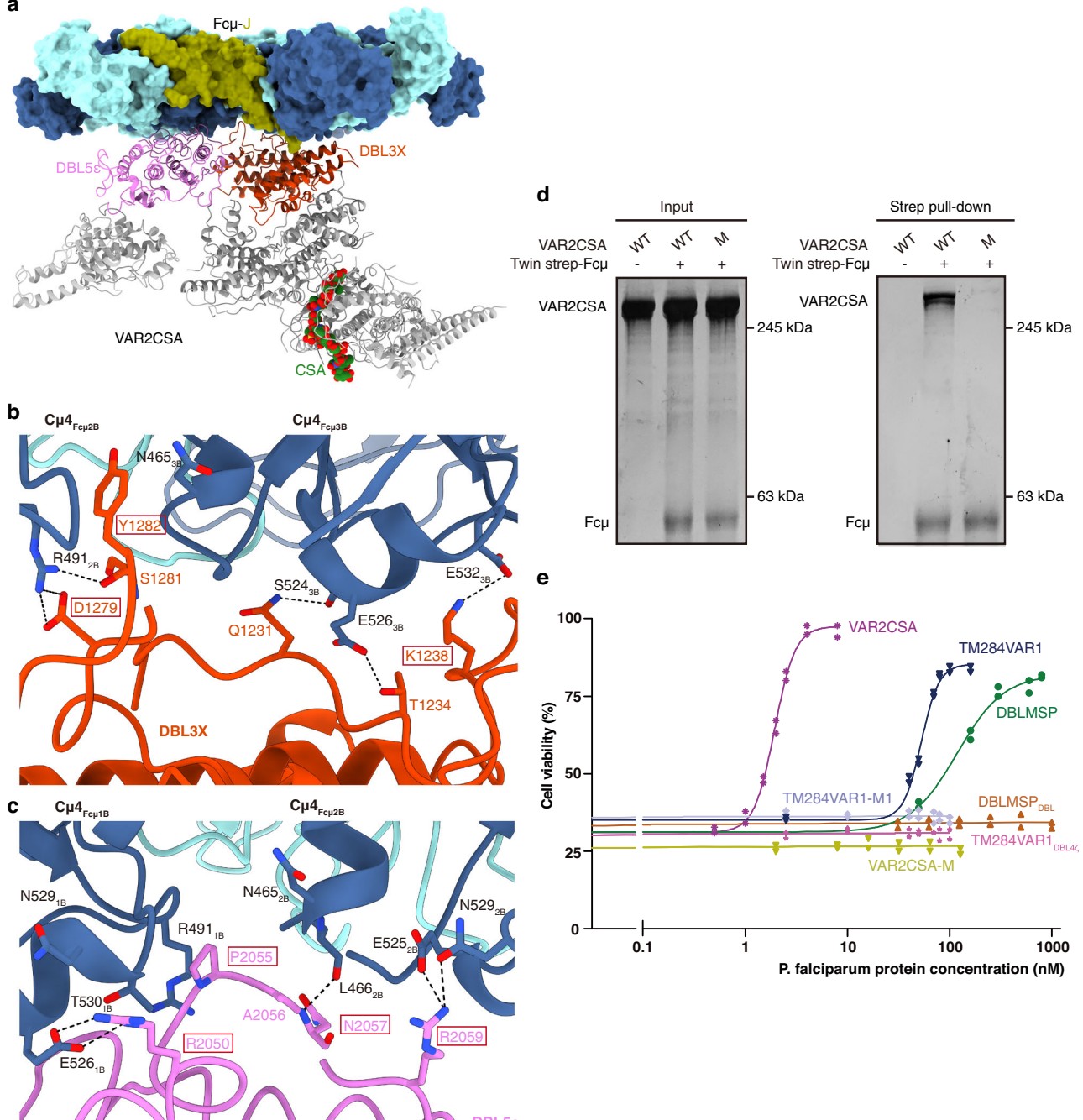

**Fig. 3 | VAR2CSA targets Fcμ via DBL3X and DBL5ε. a** A composite structural model of VAR2CSA binding to both Fcμ–J and CSA, generated by superimposing the VAR2CSA–Fcμ–J structure determined in this study to the VAR2CSA–CSA complex structure (PDB ID: 7JGH)[24]. Fcμ–J is shown using a surface representation. VAR2CSA is shown as ribbons, with the DBL3X and DBL5ε domains colored as in Fig. 2. The rest of VAR2CSA is shown in gray. CSA is shown as a space-filling model in green and red. **b** VAR2CSA_{DBL3X} interacts with the Cμ4 domains in Fcμ2B and Fcμ3B via subdomain SD1. Dashed lines indicate polar interactions. VAR2CSA residues that are mutated in the VAR2CSA-M mutant are highlighted with red boxes. **c** VAR2CSA_{DBL5ε} interacts with Fcμ1B and Fcμ2B. VAR2CSA residues that are mutated in VAR2CSA-M are highlighted with red boxes. **d** VAR2CSA-M displays

reduced binding to Fcμ–J in a pull-down experiment. For gel source data in this paper, see Supplementary Fig. 8. All pull-down experiments have been repeated three times with similar results. **e** The ectodomains of VAR2CSA and TM284VAR1, as well as full-length DBLMSP, directly inhibit IgM-mediated complement-dependent cytotoxicity. The VAR2CSA-M and TM284VAR1-M1 mutants, as well as the DBL4ζ domain of TM284VAR1 (TM284VAR1_{DBL4ζ}) and the DBL domain of DBLMSP (DBLMSP_{DBL}), display no effect. Data were analyzed by plotting the cell viabilities against the concentrations of the *P. falciparum proteins* using a 4-parameter curve-fit in the GraphPad Prism software. Two technical replicates are depicted for each experiment, and the means are used to construct the plots. Source data for two representative experiments are provided in the Source Data file.

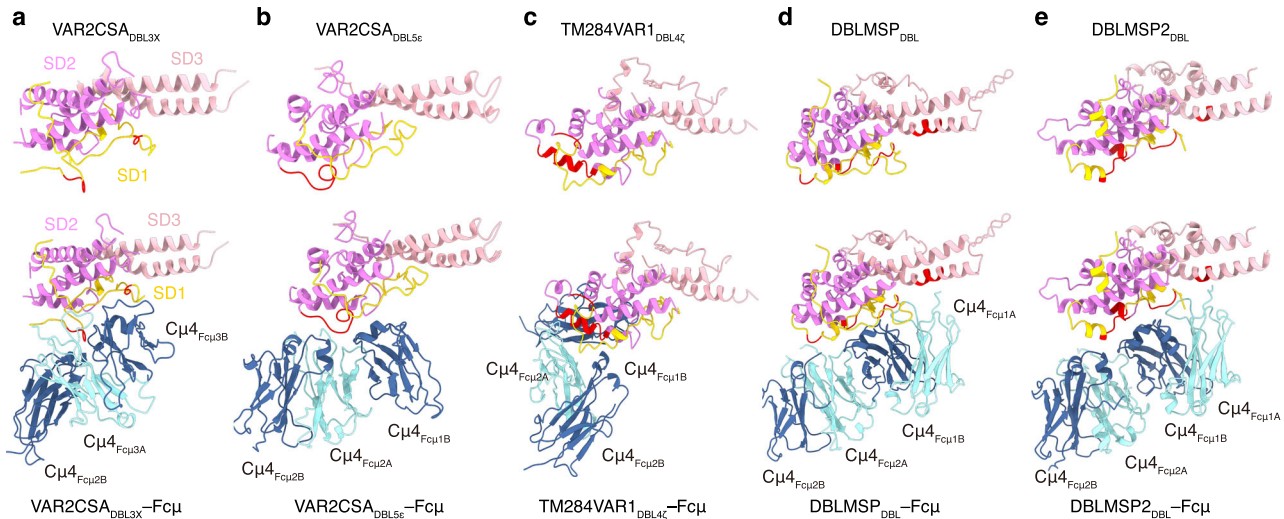

**Fig. 4 | Different binding modes between the DBL domains and Fcµ.**
**a** VAR2CSA_DBL3X structure and its complex with Fcµ. The SD1, SD2, and SD3 subdomains are colored yellow, purple, and pink, respectively. The regions involved in binding to Fcµ are highlighted in red. **b** VAR2CSA_DBL5ε structure and its complex with Fcµ. **c** TM284VAR1_DBL4ζ structure and its complex with Fcµ. **d** DBLMSP_DBL structure and its complex with Fcµ. **e** DBLMSP2_DBL structure and its complex with Fcµ.

Similar to DBLMSP_DBL, DBLMSP2_DBL also targets Fcµ1A, Fcµ1B, and Fcµ2B; and all three of its subdomains are involved in binding to these Fcµ molecules. Distinct molecular interactions are nevertheless present at the DBLMSP2_DBL–Fcµ interface when compared to that of DBLMSP_DBL–Fcµ (Fig. 4d, e). For example, several DBLMSP2 residues, including Arg214 from SD1, as well as Pro280, Thr281, and Lys283 from SD2, form extensive ionic, van der Waals, and hydrogen bond interactions with three consecutive Glu in Cµ4_Fcµ1B (Glu525–Glu527, Fig. 6e). Arg221 and Lys224 contact Glu525–Glu526 in Cµ4_Fcµ2B (Fig. 6f). These interactions are unique to the DBLMSP2_DBL–Fcµ–J complex, and likely contribute to the higher binding affinity between DBLMSP2_DBL and Fcµ–J (Fig. 1b).

## Discussion

IgM serves as the first line of defense in adaptive immunity and initiates the complement cascade to communicate with the innate immune system. The ability to hijack IgM bestows a survival advantage on *P. falciparum* and therefore can increase virulence. The four *P. falciparum* proteins investigated here interact with Fcµ using their DBL domains. In particular, VAR2CSA_DBL5ε, TM284VAR1_DBL4ζ, DBLMSP_DBL, and DBLMSP2_DBL all interact with Fcµ1–Fcµ2, and their binding sites overlap with those of pIgR/SC (Fig. 7). The unique preference of the *P. falciparum* proteins for this interaction "hot spot" could be rationalized by the asymmetrical feature of IgM. Fcµ1 exhibits more rigidity when compared to Fcµ2–Fcµ5 due to its extensive interaction with the J-chain[6]. As a result, it could be more likely to bind these *P. falciparum* proteins. Furthermore, only on this side of the Fcµ–J platform the *P. falciparum* proteins can readily interact with Fcµ2, as binding to the other side of Fcµ1 would position the Fcµ2-interacting regions of these proteins towards the gap of the IgM pentamer, where the J-chain is located instead of an Fcµ molecule. It should be noted, however, that these *P. falciparum* proteins solely target the Cµ4 domains, which are located in the structural core of the Fcµ pentamer, and are therefore expected to be relatively rigid. Thus, the precise mechanism by which *P. falciparum* proteins selectively bind to Fcµ1–Fcµ2 remains incompletely understood.

Notably, the molecular interactions between these DBL domains and Fcµ1–Fcµ2 are all different (Fig. 4). As described above, VAR2CSA_DBL5ε covers the base of Fcµ1B and Fcµ2B using subdomain SD1, and the remaining SD2–SD3 subdomains are projected toward Fcµ1. TM284VAR1_DBL4ζ mainly binds to Fcµ1–Fcµ2 using subdomain

SD2 and approaches the Fcµ plane using a completely different angle. DBLMSP_DBL and DBLMSP2_DBL bind to Fcµ1–Fcµ2 using all three subdomains; however, despite this overall similar binding pattern, each protein employs a distinct set of residues to bind these Fcµ molecules. In contrast to these four DBL domains that target Fcµ1–Fcµ2, VAR2CSA_DBL3X uniquely engages Fcµ2–Fcµ3. The DBL domains display high sequence diversity and are adaptable, interacting with a myriad of molecules[33]. It is amazing how they have evolved such diverse ways to target one human molecule. This is reminiscent of the two binding modes between the DBL domains and ICAM-1[34] and truly underscores the paramount importance of IgM for malaria immunity. It is worth noting that all of these DBL domains interact with multiple Fcµ molecules within the Fcµ pentamer; therefore, it is unlikely that they will target the monomeric form of IgM, as in the B-cell receptor complex. In fact, VAR2CSA_DBL3X, VAR2CSA_DBL5ε, and TM284VAR1_DBL4ζ all heavily exploit the space between adjacent Fcµ molecules for binding. Indeed, it has also been shown that monomeric IgM indeed does not bind to the iRBCs that display TM284VAR1 or the TM284VAR1_DBL4ζ recombinant protein[35]. On the other hand, none of them form significant interactions with the J-chain, so it is likely that they will bind well to an IgM hexamer as well.

How does the hijacking of IgM by these proteins benefit *P. falciparum* parasites? The main goal appears to be immune evasion[36]. First, IgM would facilitate the masking of antibody epitopes. A number of antibody epitopes have been identified and mapped onto the structure of VAR2CSA[24,29,37–39]. Most of the epitopes in DBL3X and DBL5ε would be concealed by Fcµ–J, such as PAM8.1, P62, P63, and P23 (Supplementary Fig. 7a). Notably, Fcµ–J only represents the IgM core. In its fully extended conformation, an entire IgM molecule containing Cµ2 and the antigen-binding fragments may reach a length of roughly 38 nm (Supplementary Fig. 7b). This substantial size renders it an optimal shelter for *P. falciparum* parasites to obstruct the binding of neutralizing antibodies. In addition, these *P. falciparum* proteins can interfere with IgM-mediated complement activation. IT4VAR60, a PfEMP1 protein, has been proposed to occupy the C1q binding site on IgM[23]. The C1q binding site is located in the Cµ3 domain of Fcµ, and involves residues 432–436 in the FG loop[40], which are positioned at the outer edge of Cµ3. However, the binding regions of the four *P. falciparum* proteins studied here are confined to the central Cµ4 domains and do not extend to the C1q binding site. We showed that the ectodomains of VAR2CSA and TM284VAR1, as well as full-length DBLMSP,

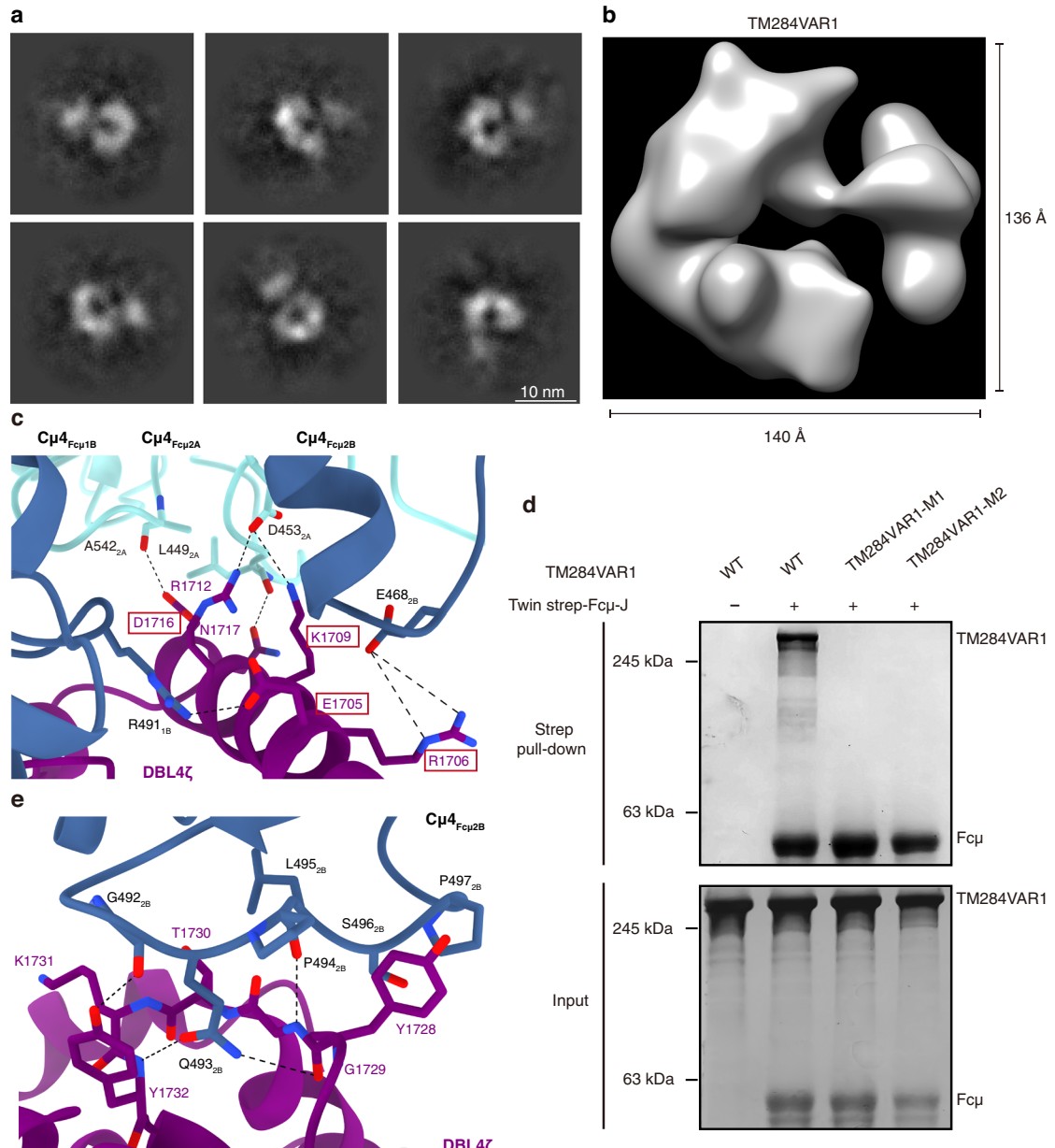

**Fig. 5 | TM284VAR1 interacts with Fcμ using the DBL4ζ domain. a** 2D classification of TM284VAR1 cryo-EM data suggests that it has a flexible structure. **b** The 3D reconstruction of TM284VAR1 at 17.8 Å. **c** Interactions between the α3 helix within SD2 of TM284VAR1$_{DBL4ζ}$ and Fcμ. TM284VAR1 residues that are mutated in TM284VAR1 mutants are highlighted with red boxes. **d** TM284VAR1 mutants display reduced binding to Fcμ–J. All pull-down experiments have been repeated three times with similar results. **e** Interactions between the Tyr1728–Tyr1732 loop of TM284VAR1$_{DBL4ζ}$ and Fcμ.

but not the monomeric TM284VAR1$_{DBL4ζ}$ and DBLMSP$_{DBL}$ domains, are capable of inhibiting complement-dependent cytotoxicity in vitro (Fig. 3e). This may be attributed to the large size of these molecules. For example, the ectodomain of VAR2CSA has a height of approximately 11.4 nm, surpassing that of the central cavity of the C1q complex, where the C1r and C1s proteases are accommodated[41] (Supplementary Fig. 7c). In vivo, these proteins are expressed on the surfaces of the iRBCs or the parasite merozoites, making it more plausible that they would directly antagonize the multivalent binding of IgM to the antigen. In any event, it is clear that these *P. falciparum* proteins can impede IgM-mediated complement activation by steric hindrance, thereby disarming another crucial component of the human immune system. Finally, these proteins can hinder the interaction between IgM and its cellular receptors. As described above, all these proteins dwell in an interaction hot spot of IgM and clearly interfere with the binding of pIgR/SC, which governs the mucosal transport of IgM. The other two IgM receptors, FcαμR and FcμR, function in the humoral immune response; and the Cμ4 domain of IgM is critical for their binding as well[35,42,43]. Indeed, the high-affinity binding site of FcμR (R1 site)[44] is also located at the aforementioned hot spot (Fig. 7). Therefore, the *P. falciparum* proteins could interfere with the perception of IgM by its receptors and suppress IgM-related immune signaling pathways.

## Methods
### Cell culture
Sf21and High Five insect cells (Invitrogen, B821-01 and B855-02) were cultured using SIM-SF and SIM-HF media (Sino Biological, MSF1 and MHF1) in a nonhumidified shaker at 27 °C. HEK293F cells (Thermo Fisher, 11625019) were cultured using SMM 293-TI medium (Sino

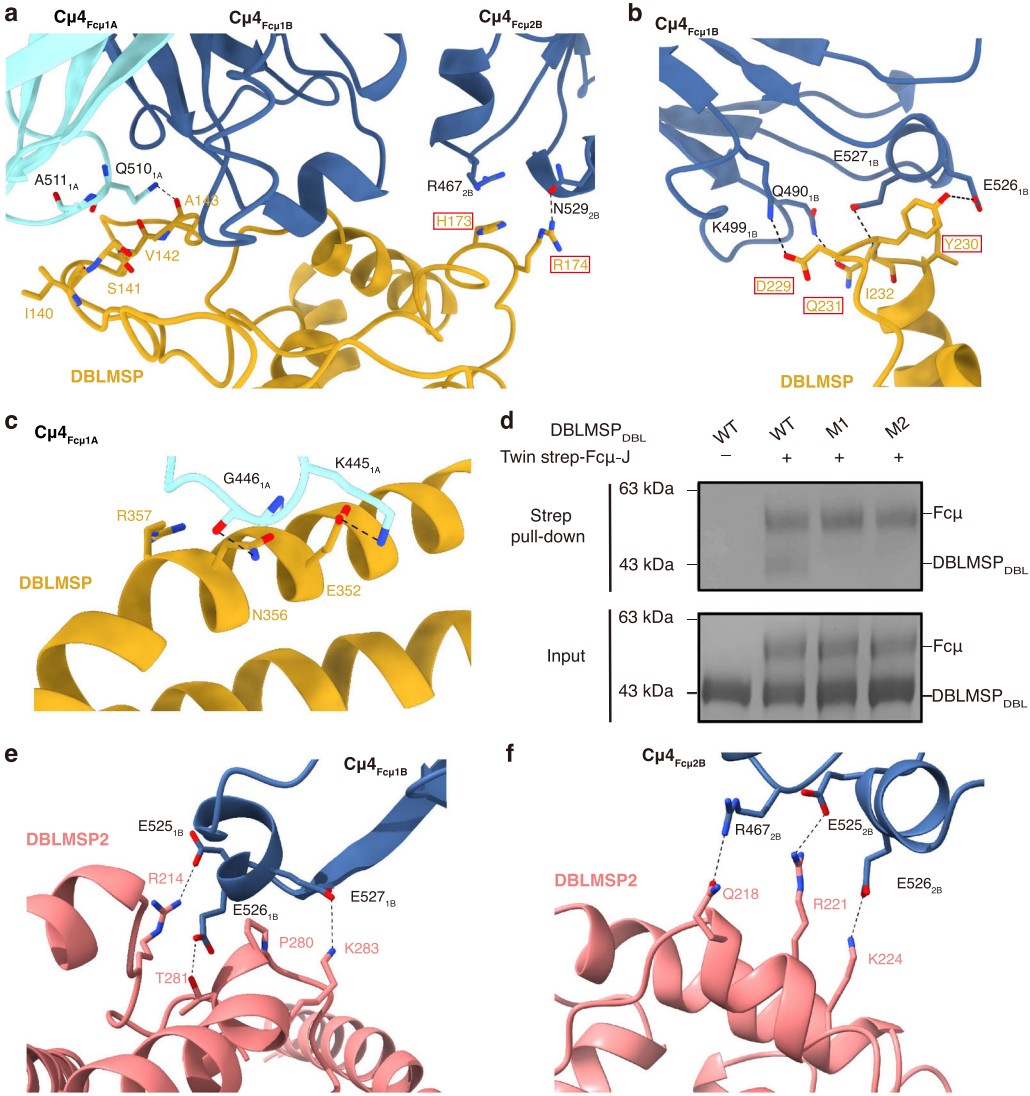

**Fig. 6 | Interactions between DBLMSP family proteins and Fcμ. a** Interactions between DBLMSP_DBL SD1 and Fcμ. DBLMSP residues that are mutated in DBLMSP-M1 are highlighted with red boxes. **b** Interactions between DBLMSP_DBL SD2 and Fcμ. DBLMSP residues that are mutated in DBLMSP-M2 are highlighted in red boxes. **c** Interactions between DBLMSP_DBL SD3 and Fcμ. **d** DBLMSP_DBL mutants display reduced binding to Fcμ-J. All pull-down experiments have been repeated three times with similar results. **e** DBLMSP2_DBL form extensive interactions with Glu525–Glu527 in Cμ4_Fcμ1B. **f** DBLMSP2_DBL interacts with Glu525–Glu526 in Cμ4_Fcμ2B.

Biological, M293TI) in a humidified shaker at 37 °C with 5% CO₂. OCI-Ly10 cells (RRID: CVCL_8795, originally purchased from the American Type Culture Collection), were cultured using RPMI-1640 (Thermo Fisher, C11875500CP) medium supplemented with 10% fetal bovine serum (PAN Seratech, ST30-3302) and 1% penicillin–streptomycin (Gibco, 15140122) in a humidified incubator at 37 °C with 5% CO₂.

**Protein expression and purification**
Codon-optimized DNAs and primers used in this study are listed in Source Data file. The DNA fragments encoding the ectodomains of VAR2CSA (PlasmoDB no. PfIT_120006100, residues 1–2599) and TM284VAR1 (Genbank no. JQ684046, residues 1–2367), as well as the DBL4ζ domain of TM284VAR1 (residues 1522–1952), were cloned into a pFastBac vector with the honeybee melittin signal peptide and a C-terminal 8×His tag. Baculoviruses were generated and amplified using Sf21 cells; and were used to infect High Five cells at a density of 1.5–2.0 million cells per mL to express the recombinant proteins. The conditioned media of High Five cells were collected 2 days after for protein purification. Codon-optimized DNAs encoding full-length

DBLMSP (Genbank no. FJ556042.1) and its DBL domain (residues 103–503), as well as the DBL domain of DBLMSP2 (PlasmoDB no. Pf3D7_1036300, residues 161–454) were cloned into a pcDNA vector with the IL-2 signal peptide and a C-terminal 8×His tag. The resulting plasmids were transfected into HEK293F cells using polyethylenimine (Polysciences, 23966-2) for protein expression. The conditioned media of transfected HEK293F cells were collected 4 days after.

For protein purification, the above High Five and HEK293F cell cultures were collected by centrifugation at 500 × g, and the conditioned media were concentrated and exchanged into Binding buffer (25 mM Tris-HCl, pH 8.0, 150 mM NaCl) using a Hydrosart Ultrafilter (Sartorius). The recombinant proteins were then isolated using the Ni-NTA affinity resin (GE healthcare, 17531803). After washing with 50 column volumes of Washing buffer (25 mM Tris-HCl, pH 8.0, 150 mM NaCl, 25 mM imidazole), the target proteins were eluted using 10 column volumes of Elution buffer (25 mM Tris-HCl, pH 8.0, 150 mM NaCl, 500 mM imidazole). Afterwards, they were further purified by size-exclusion chromatography and eluted using Binding buffer. A Superdex 6 increase column was used for VAR2CSA, TM284VAR1, and

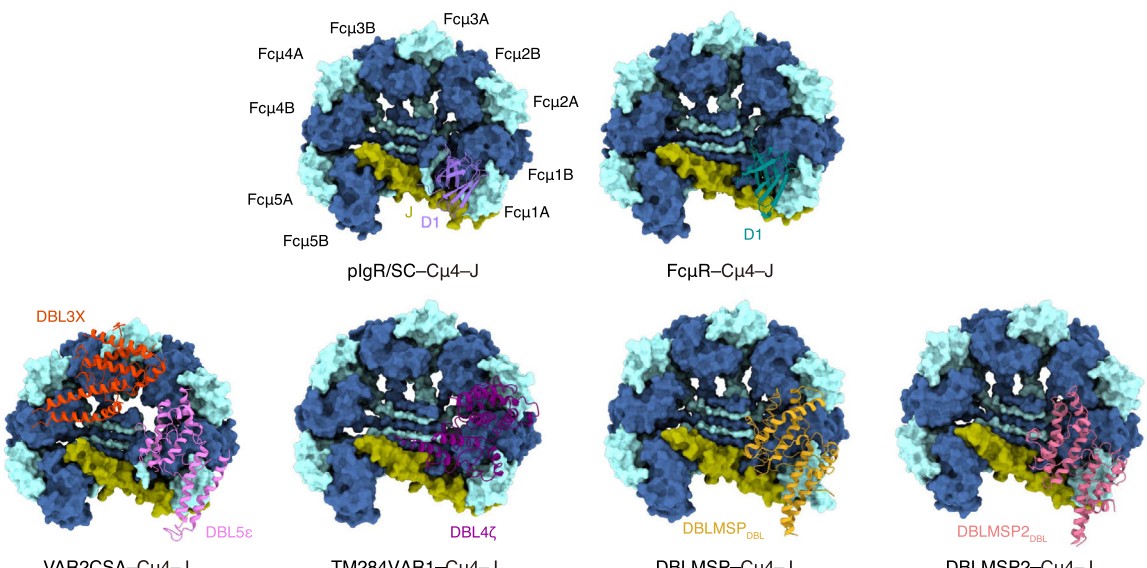

**Fig. 7 | The binding sites of VAR2CSA, TM284VAR1, DBLMSP, and DBLMSP2 on Fcμ–J overlap with that of pIgR/SC and the high-affinity site of FcμR.** The structures of Fcμ–J in complexes with the D1 domain of pIgR/SC (purple), FcμR (R1 site, green)[44], VAR2CSA$_{DBL3X–DBL5ε}$, TM284VAR1$_{DBL4ζ}$, DBLMSP$_{DBL}$, and DBLMSP$_{DBL}$ are shown in the same orientation for comparison. Fcμ–J and the DBL domains are colored as in Fig. 2. Only the Cμ4 domains of Fcμ are depicted for clarity.

full-length DBLMSP; whereas a Superdex 200 increase column was used for DBLMSP$_{DBL}$ and DBLMSP2$_{DBL}$.

Mutations were introduced into the corresponding expression plasmids using the PCR-based site-directed mutagenesis method, and the mutant proteins were purified similarly as the wildtype proteins.

To obtain the complexes formed between the *P. falciparum* proteins and Fcμ–J, purified *P. falciparum* proteins were individually mixed with Fcμ–J[4] at 2:1 molar ratios and incubated on ice for 1 h. The resulting complexes were then isolated using a Superdex 6 increase column in Final buffer (25 mM HEPES, pH 7.4, 150 mM NaCl). Protein purifications and complex assemblies were examined by reduced SDS-PAGE (4% stacking gel, and 8% or 10% separation gel) and Coomassie staining.

**Surface plasmon resonance (SPR)**
SPR experiments were performed using a Biacore T200 (GE Healthcare). 200–300 resonance units (RU) of VAR2CSA or TM284VAR1, or 1800–2000 RU of DBLMSP or DBLMSP2 was individually captured on a Series S Sensor CM5 Chip (Cytiva) in Running buffer (10 mM HEPES, pH 7.4, 0.005% (v/v) P20). Serial dilutions of purified Fcμ–J in Running buffer were then injected, ranging in concentrations from 40 nM to 2.5 nM (twofold dilutions). The SPR results were analyzed with the Biacore Evaluation Software and fitted using a 1:1 binding model.

**Cryo-EM data collection and processing**
After size-exclusion chromatography, purified ternary complexes containing Fcμ–J and the *P. falciparum* proteins were concentrated to 0.9 mg/ml. These samples were then treated with 0.05% glutaraldehyde (Sigma) at 20 °C for 10 min. The reactions were terminated by the addition of 1 M Tris-HCl (pH 7.4) to a final concentration of 100 mM. The cross-linked samples were applied onto glow-discharged holey carbon gold grids (Quantifoil, R1.2/1.3) using a Vitrobot (FEI) at 4 °C with 100% humidity. The blotting time was 0.5–1.5 s, followed by a waiting time of 5 s. The grids were then plunged into liquid ethane. Grid screenings were performed using a 200 kV Talos Arctica microscope equipped with a Ceta camera (Thermo Fisher). Data collections were performed using a 300 kV Titan Krios electron microscope (Thermo Fisher) with a K3 direct detection camera. Raw movie frames were aligned and averaged into motion-corrected summed images using MotionCor2 (v1.4.4)[45]. The contrast

transfer function (CTF) parameters were estimated using Gctf (v1.06)[46]. Subsequent data processing was carried out using cryoS-PARC (v3.2)[47] or RELION (v3.1)[48]. For the VAR2CSA–Fcμ–J sample, a total of 3,447,418 particles were initially extracted from 7,103 micrographs, which were subjected to several rounds of 2D classifications and heterogeneous refinement, resulting in 690,345 particles that were used to calculate a density map of 3.6 Å resolution. For TM284VAR1–Fcμ–J, 3,399,800 particles were extracted from 10,039 micrographs, which were subjected to 2D and 3D classifications, resulting in 849,826 particles that yielded an overall map of 3.6 Å resolution and a local map of 3.7 Å. For DBLMSP–Fcμ–J, 2,421,689 particles were extracted from 2,666 micrographs and used in classifications and refinement, resulting in 391,618 particles that yielded an overall map of 3.7 Å and a local map of 3.6 Å. For DBLMSP2–Fcμ–J, 2,777,634 particles were extracted from 3,977 micrographs and used in classifications and refinement, resulting in 458,396 good particles that yielded an overall map of 3.3 Å and a local map of 3.2 Å resolution. More details for the 3D reconstructions are presented in Supplementary Figs. 2–5. The local resolution maps were analyzed using ResMap[49] and displayed using UCSF ChimeraX[50].

**Structure building and refinement**
The cryo-EM structure of Fcμ–J (PDB ID: 6KXS)[4], the crystal structure of VAR2CSA$_{DBL3X}$ (PDB ID: 3CML)[51], and the cryo-EM structures of the ID2a–ID2b and DBL4ε–DBL5ε regions of VAR2CSA (PDB ID: 7JGE, 7JGF)[24] were docked into the EM map of the VAR2CSA–Fcμ–J complex using UCSF Chimera[50] and then adjusted using Coot[52]. Structural models of TM284VAR1$_{DBL4ζ}$ and DBLMSP$_{DBL}$ were first generated using the tFold server (https://drug.ai.tencent.com/console/en/tfold), and then fitted into EM maps and adjusted using Coot. Structural refinements were performed using real-space refinement in Phenix[53].

**Strep pull-down assay**
A twin-strep tag is present on Fcμ. Eighty micrograms of purified *P. falciparum* proteins and 40 μg of Fcμ–J proteins were incubated with StrepTactin beads (Smart Lifesciences) in Binding buffer on ice for 1 h. The beads were spun down and then washed three times using Binding buffer. Proteins retained on the beads were eluted using Binding buffer supplemented with 10 mM desthiobiotin. The results were analyzed by SDS-PAGE and Coomassie staining.

## Complement-dependent cytotoxicity assay

To produce anti-CD20 or anti-RBD IgM molecules, heavy chain DNAs of the antigen-binding fragments of rituximab or BD-368-2[54] were installed upstream of Fcμ in the pcDNA vector. The resulting anti-CD20 or anti-RBD heavy chain plasmids were transfected into HEK293F cells together with the corresponding light chain and J-chain expression plasmids using a 1:1:3 ratio. A C-terminal 8×His tag was added to the J-chain. The anti-CD20 or anti-RBD IgM proteins were then isolated from the conditioned medium using the Ni-NTA and size-exclusion chromatographies as described above. A Superdex 6 increase column and Binding buffer were used for the size-exclusion step.

The complement-dependent cytotoxicity assay was performed using OCI-Ly10 cells, which express CD20. Anti-CD20 IgM or anti-RBD IgM (1:300, 6 nM) was incubated with serially diluted *P. falciparum* proteins in 50 μL RPMI-1640 for 20 min. The resulting samples were further mixed with equal volumes of OCI-Ly10 cultures (~20,000 cells) and normal human serum complement (1:12.5 dilution, Quidel) sequentially, and then transferred into a 96-microwell plate. After 6 h of incubation at 37 °C, 50 μl of CellTiter-Glo reagent (Promega, G7572) was added to each well and incubated for 10 min at room temperature. Luminescence was measured using a Cytation 5 cell imaging multimode reader (BioTek). The data were analyzed by plotting the luminescence units against concentrations of the *P. falciparum* proteins in GraphPad Prism using a 4-parameter curve-fit.

## Reporting summary

Further information on research design is available in the Nature Portfolio Reporting Summary linked to this article.

## Data availability

Cryo-EM density maps of VAR2CSA–Fcμ–J, TM284VAR1–Fcμ–J, DBLMSP_DBL–Fcμ–J, and DBLMSP2_DBL–Fcμ–J have been deposited in the Electron Microscopy Data Bank with accession codes EMD-33542, EMD-33547, EMD-33548 (local map), EMD-33538, EMD-33539 (local map), and EMD-33805, EMD-33806 (local map), respectively. Structural coordinates have been deposited in the Protein Data Bank with accession codes 7Y0H, 7Y0J, 7Y09, and 7YG2. Previous published structural coordinates used in this study include 6KXS, 3CML, 7JGE, 7JGF, and 7JGH. Source data are provided with this paper.

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

## Acknowledgements

We thank the Core Facilities at the School of Life Sciences, Peking University for help with negative-staining EM; the Cryo-EM Platform of Peking University for help with data collection; the High-performance Computing Platform of Peking University for help with computation. We also thank the National Center for Protein Sciences at Peking University for assistance with the Biacore and BioTek Cytation Reader. Special thanks to J. Guo for insightful discussions on the complement-dependent cytotoxicity experiment. This work was supported by the Qidong-SLS Innovation Fund to J.X. and by Changping Laboratory. C.J. is supported by the Boya Postdoctoral Fellowship at Peking University.

## Author contributions

C.J. and H.S. carried out most of the experiments with the help of C.S., Y.L., and S.C. T.H.S. provided the structural model of the IgM–C1 complex. J.X. conceived and supervised the project, and wrote the manuscript with inputs from all authors.

## Competing interests

The authors declare no competing interests.
