## [Peer Review File · Nature Communications]

REVIEWER COMMENTS

Reviewer #1 (Remarks to the Author):

Binding of *Plasmodium falciparum* infected erythrocytes and invasive forms (merozoites) to human IgM is a parasite adhesion phenotype associated with severe malaria, and is thought to be an immune evasion mechanism. Several parasite proteins containing “Duffy- Binding-Like” (DBL) domains have previously been shown to bind to the Fc (C μ 4) region of IgM, but the precise details of these molecular interactions were unknown.

This manuscript provides the cryoEM structures for four *P. falciparum* DBL domain-containing proteins (two PfEMP1 variants and two merozoite proteins) binding to the human IgM Fc μ -J core protein. Although all four proteins bind to the C μ 4 region of IgM, the precise details of their binding sites differ in each case, and these differences are clearly described in the main figures and extended data. The binding sites identified from the cryoEM structures were validated by pull-down experiments using recombinant DBL proteins in which the key amino acids implicated in binding to IgM were mutated, leading to a reduction in IgM binding. The functional significance of the findings were also explored by showing that the IgM-binding DBL proteins blocked complement mediated lysis via the C1 complex, most likely due to steric effects, whereas the mutant versions of the same proteins had no inhibitory effect.

The biological significance of these findings and the possible role(s) of parasite IgM-binding in immune evasion are described in the discussion section. Overall, this work gives fascinating new insights into a malaria virulence-associated host-parasite interaction and highlights the versatility of DBL domains in their molecular interactions with host molecules and the role of IgM-binding as an important parasite immune evasion mechanism.

Comments

1) outdated nomenclature

Figure 1: some of the gene names used are out of date. While it is OK to use the old gene names in the text and figures (especially as they are widely used in prior literature), for clarity and cross-referencing, it is important to also give the current gene names (from the main *P. falciparum* genome database PlasmoDB <https://plasmodb.org>) at least once in the manuscript – either in the methods or in figure 1.

VAR2CSA from the FCR3/IT4 strain: new gene name is PfIT_120006100

DBLMSP2 from the 3D7 strain: new gene name is PF3D7_1036300

For those genes not in PlasmoDB, a Genbank or equivalent reference should be given for the gene sequence used.

Similarly, some of the nomenclature used for the DBL domain classes is out of date. Numbers are now used to indicate DBL sub-classes rather than position in the gene (see Rask et al 2010 PLoS Computational Biology and Otto et al 2019, Wellcome Open Research). i.e.

VAR2CSA domains are DBL_{pam1}-DBL_{pam2}-CIDR_{pam}-DBL_{pam3}-DBL_{epam4}-DBL_{epam5}-DBL_{ε10}.

TM284VAR1 domains are DBL_{α1.8}-CIDR_{β2}-DBL_{γ7}-DBL_{ε11}-DBL_{ζ2}-DBL_{ε6}.

As above, I think it is OK to keep the old names if the authors wish, but they should make sure that the current domain classifications are also given somewhere in the text. Maybe a short section could be added to the start of the methods describing the above gene names, Genbank links and domain classifications for the proteins studied here?

2) validation experiments with mutant proteins

Pull-down experiments showed that IgM-binding was reduced when using recombinant DBL proteins in which key amino acids identified from the cryoEM structures were mutated (Figs 3, 5 and 6). These experiments do confirm the functional importance of the binding sites identified. However, I was surprised that additional approaches such as SPR were not used (especially as SPR data for the wild type DBL domains were given earlier, Fig 1). Was there a reason why IgM-binding by SPR was not done with the mutant DBL proteins? Is IgM binding completely abolished in the mutant proteins, or do they show reduced affinity? (This may not be essential, but if the data are available, they could be added to the extended data). Figure 4 also shows that the mutant proteins have lost the ability to protect against complement-mediated lysis, providing additional validation of their reduced IgM-binding.

Another query regarding the mutant proteins is whether any measures were taken to check that the mutations did not disturb the overall structure of the protein eg. for TM284VAR1 mutants M1 and M2, did the mutations from charged residues to Alanine in the $\alpha 3$ helix of SD2 disturb this α -helix structure? Was any structural characterisation/validation of the mutant proteins carried out? (eg. circular dichroism or similar?)

Also, as noted below, some methodological details on how the mutant proteins were made is needed, and the gel purification profiles and SDS-PAGE gels of the mutant proteins should be added to extended data fig 1.

3) sparsity of methodological detail.

Many of the methods are given in a very “broad brush-stroke” manner. It would not be possible for other scientists to exactly reproduce many aspects of this work due to the lack of technical detail. Greater methodological detail in all areas would be useful (this can be given as supplementary information if needed). In particular:

i) for all reagents, plasmids and cell lines, the source should be given including both supplier and catalogue number

ii) all methods should include enough detail to allow others to reproduce the work. A few examples are – for protein expression and purification, how was buffer exchange carried out? How exactly was Ni-NTA chromatography carried out (i.e. basic details of resin used, column volume etc). Where is information on how the mutant proteins were made? Also there seem to be no details on how the DBLMSP2 protein and the TM284var1DBL4ζ protein were made, which should be included (included specific amino acids expressed).

Some basic details on SDS-PAGE should also be provided (type of gel and buffer used, reducing agent or not?) How was the alignment in extended fig 5 generated? How was the composite model in extended fig 7 generated?

In the extended data table 2, for MSPDBL2 the amino acid sequence is given rather than the codon optimised DNA sequence. It would be useful to give both the codon optimised DNA sequence and the amino acid sequence for all the proteins generated in this work.

4) discussion points

Most of the discussion points given on the functional significance of IgM binding in parasite immune evasion are plausible and interesting. However, the “demobilizing immune IgM molecules” against the parasite and preventing them from binding to their specific targets argument (page 9 lines 11-13) seems weak. The total concentration of IgM in plasma is usually ~1 mg/ml or higher, and non-parasite-specific IgM would likely always be in excess over parasite-specific IgM. How then would the parasite DBL domains preferentially bind the IgM molecules that specifically target parasite antigens? If the authors can suggest a plausible mechanism for this, it could be explored in the discussion, but if not, perhaps this discussion point should be removed?

Also in the discussion, the authors predict that because the DBL domains interact with multiple Fcμ components, it is unlikely that they would be able to bind monomeric IgM. For TM284VAR1, it has been shown experimentally that monomeric IgM does not bind to TM284VAR1-expressing infected erythrocytes nor to TM284VAR1DBL4μ recombinant protein (Ghumra et al J Immunol 2008, reference 19). So, the authors may wish to cite this publication while discussing this point, to support their argument.

Minor comments

The word “hijack” is overused, especially as it is a descriptive rather than a scientific term. It is in the title, the abstract, the final paragraph of the introduction, the first paragraph of the results and twice in the discussion. Maybe keep it in the title and discussion, but replace other uses with the simpler, more accurate “bind”?

Occasional unusual wording eg. “Structural analyses suggest that VAR2CSA occludes the congregation of the complement C1 complex on IgM”. Perhaps “prevents (or inhibits) the assembly of the complement C1 complex on IgM” would be clearer?

Fig 3 legend: “VAR2CSA targets Fc μ via DBL3X and DBL5 ϵ and inhibits IgM-mediated complement activation”. The second half of this title should be removed because the ability of VAR2CSA to inhibit IgM-mediated complement activation is not shown in this figure.

Fig 4A: typo “protein concertration”

Fig 4B: I presume this is a model based on the known structure of the C1 complex plus the structures provided in this manuscript (rather than being based on experimental data of a complex of all the proteins shown in the figure?) Please clarify in the figure legend that this is a model (if so) and give information in the methods as to how this model was derived.

Maybe remove “CDC” as an acronym throughout to improve the readability of the text? (There is already a lot of essential clunky nomenclature here, so why add to it unnecessarily?)

Typo: Page 11 line 10 should be TM284VAR1 not TM284var1.

Extended data Fig 5: on my screen it was hard to read the TM284VAR1 amino acids covered by the purple highlighting – maybe a paler colour could be used?

Also, some of the amino acid numbering and residues highlighted here did not seem to match the main text. For example, for VAR2CSA DBL5 ϵ , the text identifies R2050 P2055 A2056 R2059 as the main interacting residues, but R2059 is not highlighted on the figure, and the numbering of the amino acids does not match that given in the text. Please check and correct all highlighting and numbering in this figure.

Reviewer #2 (Remarks to the Author):

Summary paragraph

In the manuscript by Ji et al., authors characterize recognition of IgM molecule by four Plasmodium falciparum proteins from PfEMP1 family, including VAR2CSA, TM284VAR1, DBLMSP and DBLMSP2. By utilizing cryoEM authors were able to provide molecular details of IgM targeting by this group of virulent proteins, which contribute to malarial immune evasion. Interestingly, even though all proteins interfere with IgM-mediated complement activation, they present different modes of interaction with IgM molecules. This manuscript provides insight into different modes of IgM sequestration by important human parasite, and therefore warrants a publication in Nature Communication journal, with adjustments outlined below:

Major points:

-function and significance of VAR2CSA is described in detail in the introduction, but information about DBLMSP, DBLMSP2 and TM284VAR1 is sparse. Authors should explain in more detail why those three proteins were selected for structural characterization with IgM (are they the only members of PfEMP1 family that interact with IgM? Are they significantly contributing to virulence?). TM284VAR1 is not even introduced till the results section.

-side-chain outliers for model 7YG2 are 38.9%. Values below 1% are ideal. Please correct. Fixing those outliers is probably going to slightly affect the binding interface, so authors should re-evaluate it after correcting.

Minor points:

-I don't understand this sentence in section 30 of Introduction: "In contrast to the PfEMP1 proteins that reside on iRBCs, DBLMSP and DBLMSP2 are located on the surface of P. falciparum merozoites, which are directly exposed to humoral immunity" – aren't the PfEMP1 proteins on iRBCs continuously exposed to antibodies while merozoites are hidden within iRBCs for most of their life cycle?

-Figures are poorly annotated. For example, there is no explanation of why some residues are highlighted with red boxes in Fig. 3b, c and other figures. In Fig. 4b authors show models of C1q, C1r, C1s, but there is no reference as to where those models are coming from (like PDB ID).

-why wasn't DBLMSP2 tested in CDC assay just like VAR2CSA, TM284VAR1, and DBLMSP?

-constructs used in the study (Fig. 1a) should be shown in the context of the full-length protein. It would also be useful if authors could highlight how much of a given construct they were able to resolve in their cryoEM maps.

-there should be some confirmation of proper folding of VAR2CSA-M mutant, perhaps SEC profile. Same for other proteins that contain multiple point mutations.

-Was chondroitin sulfate A added to the VAR2CSA-Fc μ -J complex? The molecule is visible in Fig. 3a but there is no explanation in the methods.

-from supplementary figures and methods section it is not clear which step of cryoEM processing was performed using which software (Relion vs. cryoSPARC).

-due to the resolution of 3.6-3.7 Å for most of the models and "bulkiness" of some side-chain densities, authors should re-consider discussing hydrogen bonds in their models. I did not examine the maps in details but I don't believe the confidence in placement of the side-chains is that high. The clash-score is also high for all models, with significant overlap between some residues.

- TM284VAR1-map looks very over-refined, and based on the 2D classes I doubt the 8Å estimate. Using soft, static mask with lots of padding in NU refinement might help in obtaining more realistic map.

Reviewer #3 (Remarks to the Author):

In this manuscript, the authors report the interactions with IgM of four plasmodium proteins containing Duffy-binding-like domains (DBL) based on cryo-EM analysis of complexes. This is an advance on previous studies of one of the proteins alone (VAR2CSA) and the three additional proteins are solved de novo. The interactions studied in this paper are important to an understanding of the immune response to infection by plasmodium. It is a comprehensive study, with detailed analysis of the binding interfaces and confirmed by mutagenesis of binding site residues. The authors discuss the functional importance of IgM binding to plasmodium infection, consistent with functions in shielding from antibody and complement recognition. Ideally, more conclusions may be drawn on general principles of DBL domain binding which will be of interest to a wider audience.

Major concerns:

1. There are potentially 10 binding sites on IgM. Why do all four proteins bind to the subunit Fc μ 1-2 position? This could be explained by interaction with asymmetric features such as the J-chain or tailpieces, but such interactions are not specifically described in the ms. Maps in validation report (Section 9.1) for TM284VAR1 appear to suggest an additional binding site.

2. Experiments in the paper suggest that the plasmodium proteins have an inhibitory effect on the complement pathway. In figure 4, a model is proposed whereby plasmodium proteins sterically inhibit complement, which may explain the assay results. However, the plasmodium proteins are membrane associated. When binding IgM, the C1q binding sites on the opposite side which face away from the membrane are available for complement deposition. In principle, the mechanism of complement inhibition could be blocking IgM multivalent binding of parasite surface antigens rather than the mechanism proposed by the authors.

3. The structural models may need further refinement. This is important to analysis of the interactions, the main point of the paper. Values in the structure data table do not generally match those in the validation reports. DBLMSP2 has significant side-chain outliers. Please report map-model FSC values. A measure of particle distribution or anisotropy of the maps is recommended. In addition, the model TM284VAR1 may not fit the map very well, based on atom inclusion scores, Q-scores, and map-model overlay in the validation report. These should be examined. Validation reports may be missing for EMD-33548, 33539, 33806.

Additional specific points:

1. Differences in the structures of the free protein compared to the bound protein would be important to compare (e.g. VAR2CSA).

2. The authors may consider reorganising the figures and panels to make them more compatible with the flow of the text. For example, the subdomains (SD1-3) are discussed intensively in the text, but not

shown until the last figure. Separate overview models of each molecule (Fig. 2) could be in the same figure as the detailed interaction pictures (e.g. Fig. 5 and 6). This is more successful in Figure 3.

3. There is no introductory material for the TM284VAR1 protein.

4. Figure 1 requires more extensive labelling including colours in Figure 1b.

5. Figure 3 title includes complement activation which is not part of the figure content.

REVIEWER COMMENTS

Reviewer #1 (Remarks to the Author):

Binding of *Plasmodium falciparum* infected erythrocytes and invasive forms (merozoites) to human IgM is a parasite adhesion phenotype associated with severe malaria, and is thought to be an immune evasion mechanism. Several parasite proteins containing “Duffy- Binding-Like” (DBL) domains have previously been shown to bind to the Fc (C μ 4) region of IgM, but the precise details of these molecular interactions were unknown.

This manuscript provides the cryoEM structures for four *P. falciparum* DBL domain-containing proteins (two PfEMP1 variants and two merozoite proteins) binding to the human IgM Fc μ -J core protein. Although all four proteins bind to the C μ 4 region of IgM, the precise details of their binding sites differ in each case, and these differences are clearly described in the main figures and Supplementary. The binding sites identified from the cryoEM structures were validated by pull-down experiments using recombinant DBL proteins in which the key amino acids implicated in binding to IgM were mutated, leading to a reduction in IgM binding. The functional significance of the findings were also explored by showing that the IgM-binding DBL proteins blocked complement mediated lysis via the C1 complex, most likely due to steric effects, whereas the mutant versions of the same proteins had no inhibitory effect.

The biological significance of these findings and the possible role(s) of parasite IgM-binding in immune evasion are described in the discussion section. Overall, this work gives fascinating new insights into a malaria virulence-associated host-parasite interaction and highlights the versatility of DBL domains in their molecular interactions with host molecules and the role of IgM-binding as an important parasite immune evasion mechanism.

Comments

1) outdated nomenclature

Figure 1: some of the gene names used are out of date. While it is OK to use the old gene names in the text and figures (especially as they are widely used in prior literature), for clarity and cross-referencing, it is important to also give the current gene names (from the main *P. falciparum* genome database PlasmoDB <https://plasmodb.org>) at least once in the manuscript – either in the methods or in figure 1.

VAR2CSA from the FCR3/IT4 strain: new gene name is PfIT_120006100

DBLMSP2 from the 3D7 strain: new gene name is PF3D7_1036300

For those genes not in PlasmoDB, a Genbank or equivalent reference should be given for the gene sequence used.

Similarly, some of the nomenclature used for the DBL domain classes is out of date. Numbers are now used to indicate DBL sub-classes rather than position in the gene (see Rask et al 2010 PLoS Computational Biology and Otto et al 2019, Wellcome Open Research). i.e.

VAR2CSA domains are DBL μ 1-DBL μ 2-CIDR μ -DBL μ 3-DBL ϵ 4-DBL ϵ 5-DBL ϵ 10.

TM284VAR1 domains are DBL α 1.8-CIDR β 2-DBL γ 7-DBL ϵ 11-DBL ζ 2-DBL ϵ 6.

As above, I think it is OK to keep the old names if the authors wish, but they should make sure that the current domain classifications are also given somewhere in the text. Maybe a short section could be added to the start of the methods describing the above gene names, Genbank links and domain classifications for the proteins studied here?

We thank the the reviewer for the positive comments and insightful suggestions. The new gene

name from database PlasmoDB or Genebank ID is included in the revised Methods. A short section has also been added to the legend of Fig. 1a to describe the old and current domain classifications for the *P. falciparum* proteins studied here.

2) validation experiments with mutant proteins

Pull-down experiments showed that IgM-binding was reduced when using recombinant DBL proteins in which key amino acids identified from the cryoEM structures were mutated (Figs 3, 5 and 6). These experiments do confirm the functional importance of the binding sites identified. However, I was surprised that additional approaches such as SPR were not used (especially as SPR data for the wild type DBL domains were given earlier, Fig 1). Was there a reason why IgM-binding by SPR was not done with the mutant DBL proteins? Is IgM binding completely abolished in the mutant proteins, or do they show reduced affinity? (This may not be essential, but if the data are available, they could be added to the Supplementary). Figure 4 also shows that the mutant proteins have lost the ability to protect against complement-mediated lysis, providing additional validation of their reduced IgM-binding.

We did test the binding between a TM284VAR1 mutant and IgM using SPR; nevertheless, its interaction with IgM became so weak, and the RU response was so low that we could not reliably deduce the K_d value. Therefore, we did not further pursue SPR measurements for the mutants. The mutant proteins display abolished binding to IgM in pull-down assays, and lost their abilities to inhibit IgM-mediated complement-dependent cytotoxicity. These results validate our structural analyses.

Another query regarding the mutant proteins is whether any measures were taken to check that the mutations did not disturb the overall structure of the protein eg. for TM284VAR1 mutants M1 and M2, did the mutations from charged residues to Alanine in the $\alpha 3$ helix of SD2 disturb this α -helix structure? Was any structural characterisation/validation of the mutant proteins carried out? (eg. circular dichroism or similar?)

Also, as noted below, some methodological details on how the mutant proteins were made is needed, and the gel purification profiles and SDS-PAGE gels of the mutant proteins should be added to Supplementary fig 1.

The mutant proteins were purified well and eluted as monodisperse peaks on the gel filtration columns, suggesting that the overall structures of these proteins are not disrupted. We have included the gel filtration profiles and SDS-PAGE analyses of the five mutants proteins in this study in revised Supplementary Fig. 1f-j. Relevant methodological details have also been described in the revised Methods accordingly.

3) sparsity of methodological detail.

Many of the methods are given in a very “broad brush-stroke” manner. It would not be possible for other scientists to exactly reproduce many aspects of this work due to the lack of technical detail. Greater methodological detail in all areas would be useful (this can be given as supplementary information if needed). In particular:

- i) for all reagents, plasmids and cell lines, the source should be given including both supplier and catalogue number
- ii) all methods should include enough detail to allow others to reproduce the work. A few examples are – for protein expression and purification, how was buffer exchange carried out? How exactly was Ni-NTA chromatography carried out (i.e. basic details of resin used, column volume etc). Where is information on how the mutant proteins were made? Also there seem to be no details on how the DBLMSP2 protein and the TM284var1DBL4 ζ protein were made, which

should be included (included specific amino acids expressed).

Some basic details on SDS-PAGE should also be provided (type of gel and buffer used, reducing agent or not?) How was the alignment in extended fig 5 generated? How was the composite model in extended fig 7 generated?

In the Supplementary table 2, for MSPDBL2 the amino acid sequence is given rather than the codon optimised DNA sequence. It would be useful to give both the codon optimised DNA sequence and the amino acid sequence for all the proteins generated in this work.

We have expanded the Methods section and included more details accordingly. Both the codon optimized DNA sequence and the amino acid sequence are included in the revised Supplementary Table 2.

4) discussion points

Most of the discussion points given on the functional significance of IgM binding in parasite immune evasion are plausible and interesting. However, the “demobilizing immune IgM molecules” against the parasite and preventing them from binding to their specific targets argument (page 9 lines 11-13) seems weak. The total concentration of IgM in plasma is usually ~1 mg/ml or higher, and non-parasite-specific IgM would likely always be in excess over parasite-specific IgM. How then would the parasite DBL domains preferentially bind the IgM molecules that specifically target parasite antigens? If the authors can suggest a plausible mechanism for this, it could be explored in the discussion, but if not, perhaps this discussion point should be removed?

We thank the reviewer for this excellent point and have removed this discussion point in the revision as suggested.

Also in the discussion, the authors predict that because the DBL domains interact with multiple Fc μ components, it is unlikely that they would be able to bind monomeric IgM. For TM284VAR1, it has been shown experimentally that monomeric IgM does not bind to TM284VAR1-expressing infected erythrocytes nor to TM284VAR1DBL4 μ recombinant protein (Ghumra et al J Immunol 2008, reference 19). So, the authors may wish to cite this publication while discussing this point, to support their argument.

We have included this discussion point as suggested.

Minor comments

The word “hijack” is overused, especially as it is a descriptive rather than a scientific term. It is in the title, the abstract, the final paragraph of the introduction, the first paragraph of the results and twice in the discussion. Maybe keep it in the title and discussion, but replace other uses with the simpler, more accurate “bind”?

We have only kept “hijack” in the title and discussion, and replaced it with “bind” or “interact with” in all other places.

Occasional unusual wording eg. “Structural analyses suggest that VAR2CSA occludes the congregation of the complement C1 complex on IgM”. Perhaps “prevents (or inhibits) the assembly of the complement C1 complex on IgM” would be clearer?

We have changed “occlude the congregation” to “prevent the assembly”.

Fig 3 legend: "VAR2CSA targets Fc μ via DBL3X and DBL5 ϵ and inhibits IgM-mediated complement activation". The second half of this title should be removed because the ability of VAR2CSA to inhibit IgM-mediated complement activation is not shown in this figure.

We have removed the second half in the title as suggested.

Fig 4A: typo "protein concerntration"

We apologize for the typo and made the correction accordingly.

Fig 4B: I presume this is a model based on the known structure of the C1 complex plus the structures provided in this manuscript (rather than being based on experimental data of a complex of all the proteins shown in the figure?) Please clarify in the figure legend that this is a model (if so) and give information in the methods as to how this model was derived.

We have provied more details in the figure legend.

Maybe remove "CDC" as an acronym throughout to improve the readability of the text? (There is already a lot of essential clunky nomenclature here, so why add to it unnecessarily?)

We have removed this acronym throughout as suggested.

Typo: Page 11 line 10 should be TM284VAR1 not TM284var1.

We have made the change as suggested.

Supplementary Fig 5: on my screen it was hard to read the TM284VAR1 amino acids covered by the purple highlighting – maybe a paler colour could be used?

Paler color was used to make words clearer.

Also, some of the amino acid numbering and residues highlighted here did not seem to match the main text. For example, for VAR2CSA DBL5 ϵ , the text identifies R2050 P2055 A2056 R2059 as the main interacting residues, but R2059 is not highlighted on the figure, and the numbering of the amino acids does not match that given in the text. Please check and correct all highlighting and numbering in this figure.

We apologized for this error and made corrections accordingly. The residues highlighted on the figure using red boxes are residues that are mutated in the VAR2CSA-M mutant.

Reviewer #2 (Remarks to the Author):

Summary paragraph

In the manuscript by Ji et al., authors characterize recognition of IgM molecule by four Plasmodium falciparum proteins from PfEMP1 family, including VAR2CSA, TM284VAR1, DBLMSP and DBLMSP2. By utilizing cryoEM authors were able to provide molecular details of IgM targeting by this group of virulent proteins, which contribute to malarial immune evasion. Interestingly, even though all proteins interfere with IgM-mediated complement activation, they present different modes of interaction with IgM molecules. This manuscript provides insight into different modes of IgM sequestration by important human parasite, and therefore warrants a publication in Nature Communication journal, with adjustments outlined below:

Major points:

-function and significance of VAR2CSA is described in detail in the introduction, but information about DBLMSP, DBLMSP2 and TM284VAR1 is sparse. Authors should explain in more detail why those three proteins were selected for structural characterization with IgM (are they the only members of PfEMP1 family that interact with IgM? Are they significantly contributing to virulence?). TM284VAR1 is not even introduced till the results section.

We have included more descriptions of these proteins in the revised Introduction.

-side-chain outliers for model 7YG2 are 38.9%. Values below 1% are ideal. Please correct. Fixing those outliers is probably going to slightly affect the binding interface, so authors should re-evaluate it after correcting.

We apologize for this, and have further refined this structure. The refined model has 1% side-chain outliers, as shown in the corresponding wwPDB validation report. We want to emphasize that the poorly refined residues in the previous model were not located in the binding interface, so our descriptions about the protein-protein interaction in this structure remain correct.

Minor points:

-I don't understand this sentence in section 30 of Introduction: "In contrast to the PfEMP1 proteins that reside on iRBCs, DBLMSP and DBLMSP2 are located on the surface of P. falciparum merozoites, which are directly exposed to humoral immunity" – aren't the PfEMP1 proteins on iRBCs continuously exposed to antibodies while merozoites are hidden within iRBCs for most of their life cycle?

We have consolidated the introduction part for the DBLMSP proteins and removed "which are directly exposed to humoral immunity" subclause that causes confusion.

-Figures are poorly annotated. For example, there is no explanation of why some residues are highlighted with red boxes in Fig. 3b, c and other figures. In Fig. 4b authors show models of C1q, C1r, C1s, but there is no reference as to where those models are coming from (like PDB ID).

Residues that are mutated in the mutant proteins are highlighted with red boxes. We have explained this in the corresponding figure legends. The C1-IgM complex structural model is provided by Dr. Thomas H. Sharp, one of the coauthors of this paper; and the corresponding reference is provided in the revised legend.

-why wasn't DBLMSP2 tested in CDC assay just like VAR2CSA, TM284VAR1, and DBLMSP?

We only acquired DNA for the DBLMSP2_{DBL} domain, but not full-length DBLMSP2, when we carried out the study, so we were not able to test DBLMSP2 in CDC assay at the time. After the structure was solved, we noticed that DBLMSP2_{DBL} occupies a very similar region on the IgM platform as DBLMSP_{DBL} (Fig. 2c and 2d), although some detailed molecular interactions are different. DBLMSP2 is also highly similar to DBLMSP1 in domain organization, containing a SPAM domain that is responsible for oligomer formation and also required for CDC inhibition (Fig. 1a, 5a). Based on these similarities, we think it is highly likely that DBLMSP2 would inhibit IgM-mediated complement activation in a manner similar to DBLMSP.

-constructs used in the study (Fig. 1a) should be shown in the context of the full-length protein. It would also be useful if authors could highlight how much of a given construct they were able to resolve in their cryoEM maps.

We thank the reviewer for this suggestion. In the revised Fig. 1a, black lines were used indicate protein fragments that are recombinantly produced in this study, whereas red arrows were used to indicate regions that are structurally modeled into the density maps.

-there should be some confirmation of proper folding of VAR2CSA-M mutant, perhaps SEC profile. Same for other proteins that contain multiple point mutations.

We have provided the SEC profiles for all the mutant proteins in the revised Supplementary Fig. 1. The mutant proteins were purified well and eluted as monodisperse peaks on the size exclusion columns, suggesting that the overall structures of these proteins are not disrupted.

-Was chondroitin sulfate A added to the VAR2CSA-Fc μ -J complex? The molecule is visible in Fig. 3a but there is no explanation in the methods.

We apologize for the confusion. CSA is not present in our structure of the VAR2CSA-Fc μ -J complex. Fig. 3a represents a composite structural model generated by superimposing our structure to the VAR2CSA-CSA structure (PDB ID: 7JGH). We have clarified this in the revised figure legend.

-from supplementary figures and methods section it is not clear which step of cryoEM processing was performed using which software (Relion vs. cryoSPARC).

Relion was only used in the TM284VAR1 structure. We have clearly indicated where these softwares were used in the revised Supplementary Fig. 2-5, and.

-due to the resolution of 3.6-3.7 Å for most of the models and "bulkiness" of some side-chain densities, authors should re-consider discussing hydrogen bonds in their models. I did not examine the maps in details but I don't believe the confidence in placement of the side-chains is that high. The clash-score is also high for all models, with significant overlap between some residues.

We have further refined our structures to improve the geometry and clash-score, etc. The "bad" residues are all located outside the binding interface, so our main conclusions remain unaffected. In particular, we cross-examined the structural models and EM density maps to ensure that the

amino acid side chains we described are of high confidence. In fact, the density maps of these residues are shown in the Supplementary figures.

- TM284VAR1-map looks very over-refined, and based on the 2D classes I doubt the 8Å estimate. Using soft, static mask with lots of padding in NU refinement might help in obtaining more realistic map.

We performed the NU refinement with soft, static mask as suggested to obtain a more realistic map. Indeed, the resolution estimate is lower (~18 Å), but the overall shape of TM284VAR1 remains unaltered. This change only affects the TM284VAR1 alone map, and would not affect our analyses of the TM284VAR1-Fc μ -J complex structure.

Reviewer #3 (Remarks to the Author):

In this manuscript, the authors report the interactions with IgM of four plasmodium proteins containing Duffy-binding-like domains (DBL) based on cryo-EM analysis of complexes. This is an advance on previous studies of one of the proteins alone (VAR2CSA) and the three additional proteins are solved de novo. The interactions studied in this paper are important to an understanding of the immune response to infection by plasmodium. It is a comprehensive study, with detailed analysis of the binding interfaces and confirmed by mutagenesis of binding site residues. The authors discuss the functional importance of IgM binding to plasmodium infection, consistent with functions in shielding from antibody and complement recognition. Ideally, more conclusions may be drawn on general principles of DBL domain binding which will be of interest to a wider audience.

Major concerns:

1. There are potentially 10 binding sites on IgM. Why do all four proteins bind to the subunit Fc μ 1-2 position? This could be explained by interaction with asymmetric features such as the J-chain or tailpieces, but such interactions are not specifically described in the ms. Maps in validation report (Section 9.1) for TM284VAR1 appear to suggest an additional binding site.

We thank the reviewer for this excellent point, and included relevant discussions in the revised manuscript: "The unique preference of the *P. falciparum* proteins for this interaction "hot spot" can be rationalized by the asymmetrical feature of IgM. Fc μ 1 exhibits more rigidity when compared to Fc μ 2-5 due to its extensive interaction with the J-chain⁶, and therefore is likely easier to be approached by the *P. falciparum* proteins. Furthermore, only on this side of the Fc μ -J platform the *P. falciparum* proteins can readily interact with Fc μ 2 as well. If they choose to land on the other side of Fc μ 1, their Fc μ 2-interacting regions would be projected towards the gap of the IgM pentamer, where J-chain resides in place of an Fc μ molecule." (Discussion, 1st paragraph)

Densities for the TM284VAR1 molecule on the other side are very weak and could not be unambiguously analyzed, since the TM284VAR1 molecule here is not tightly bound as discussed above. To be accurate, we did make changes to the general cryo-EM structure determination section in the revised paper: "Although some PfEMP1 proteins can bind IgM in a 2:1 ratio^{22,23}, 1:1 complexes were most clearly resolved for the four *P. falciparum* proteins investigated in this study." (Page 4, lines 30-31)

2. Experiments in the paper suggest that the plasmodium proteins have an inhibitory effect on the complement pathway. In figure 4, a model is proposed whereby plasmodium proteins sterically inhibit complement, which may explain the assay results. However, the plasmodium proteins are membrane associated. When binding IgM, the C1q binding sites on the opposite side which face away from the membrane are available for complement deposition. In principle, the mechanism of complement inhibition could be blocking IgM multivalent binding of parasite surface antigens rather than the mechanism proposed by the authors.

We thank the reviewer again for this excellent comment. We agree with the reviewer this could be another potential mechanism, and included this point in the revised paper: "Alternatively, the large VAR2CSA could also directly antagonize the multivalent binding of IgM to its antigen, especially when it is displayed on the surface of iRBCs." (Page 7, lines 7-9)

3. The structural models may need further refinement. This is important to analysis of the

interactions, the main point of the paper. Values in the structure data table do not generally match those in the validation reports. DBLMSP2 has significant side-chain outliers. Please report map-model FSC values. A measure of particle distribution or anisotropy of the maps is recommended. In addition, the model TM284VAR1 may not fit the map very well, based on atom inclusion scores, Q-scores, and map-model overlay in the validation report. These should be examined. Validation reports may be missing for EMD-33548, 33539, 33806.

We further refined the structural models. Angular distribution of the EM particles, as well as model-to-map FSC curves, are present in the revised Supplementary Figures 2-5. Validation reports are also provided accordingly. We want to emphasize that the poorly refined residues in the previous models were not located in the binding interface, so our descriptions about the protein-protein interaction in this structure remain correct. We cross-examined the structural models and EM density maps to ensure that the amino acid side chains in the interfaces are of high confidence. In fact, the density maps of these residues are shown in the Supplementary Figures 2-5.

Additional specific points: 1. Differences in the structures of the free protein compared to the bound protein would be important to compare (e.g. VAR2CSA).

We have included a new figure to compare the structural differences of VAR2CSA: Supplementary Fig. 2f. Compared to the VAR2CSA structure determined in the absence of IgM, a large swing of the DBL5 ϵ -DBL6 ϵ arm can be observed. This has been included in the revised paper as well. (Page 5, lines 14-15)

2. The authors may consider reorganising the figures and panels to make them more compatible with the flow of the text. For example, the subdomains (SD1-3) are discussed intensively in the text, but not shown until the last figure. Separate overview models of each molecule (Fig. 2) could be in the same figure as the detailed interaction pictures (e.g. Fig. 5 and 6). This is more successful in Figure 3.

We made some changes to the figure arrangements as suggested to improve the text flow. The subdomains figure is now Fig. 4. Also, the original Data Fig. 4 is now Fig. 8. We kept the original Fig. 2 to allow the comparisons of how different *P. falciparum* proteins bind to IgM.

3. There is no introductory material for the TM284VAR1 protein.

We included introductions to TM284VAR1: "TM284VAR1 is a PfEMP1 protein isolated from a cerebral parasite strain¹². Like VAR2CSA, TM284VAR1 can cause rosetting, namely, the adhesion of iRBCs to uninfected RBCs. It is highly likely that TM284VAR1 contributes significantly to the virulence of this cerebral malaria strain." (Page 3, lines 26-29).

4. Figure 1 requires more extensive labelling including colours in Figure 1b.

We have made these changes as suggested.

5. Figure 3 title includes complement activation which is not part of the figure content.

This is removed from the Figure 3 title.

REVIEWER COMMENTS

Reviewer #2 (Remarks to the Author):

All reviewers' comments were addressed in the updated version of the manuscript in an appropriate manner.

minor points:

-instead of using cryoSPARC please use cryoSPARC2 or 3 (whichever version was used),

-please add [Å] symbol to all figure panels with local resolution,

-In supplementary Figure 3h there is a free-floating letter "f" in the lower left panel,

-it is sufficient to round KD, Koff and Kon results up to one decimal place (Fig.1b),

-please expand Methods section to include more details regarding cryoEM data processing, including: number of micrographs, number of particles extracted, final number of particles in reconstruction and resolution for each map, etc.

-lines 235-239: consider including changes highlighted with " "; for example using word 'choose' in reference to proteins sounds a bit strange, as they are not sentient...

Fc μ 1 exhibits more rigidity when compared to Fc μ 2-5 due to its extensive interaction with the J-chain⁶, and therefore "might be more likely to bind" *P. falciparum* proteins. Furthermore, only on this side of the Fc μ -J platform the *P. falciparum* proteins can readily interact with Fc μ 2 as well. "Binding of *P. falciparum* proteins to the other side of Fc μ 1 is unlikely, as" their Fc μ 2-interacting regions would be projected towards the gap of the IgM pentamer, where J chain resides in place of an Fc μ molecule.

Reviewer #3 (Remarks to the Author):

The revised ms is improved in presentation and the overall quality of the structural models obtained from cryo-EM data is now satisfactory. The results will be of interest to the wide audience of Nature Communications. However, there is still a major issue that needs to be addressed by the authors because the mechanism for inhibiting complement protein activation is still in question.

1. Major: The authors should address the serious concern about the mechanism proposed for complement deposition and activation that was raised during the first round of review. The protein ectodomains were expressed as soluble molecules for structural study and interfere with in vitro complement activation in the assay as reported in Figure 5a, and Figure 5b illustrates a structural interpretation of the binding assay which we agree with. In vivo, however, the proteins are associated with the surface of either the merozoite or the infected blood cell membrane. Therefore, when binding IgM, they will not be available to block complement binding at the same time. The interpretation of the structures and assays cannot, without additional considerations, reflect their role in vivo. We suggested an alternative different mechanism which the authors have included, but the incorrect mechanism should be removed from the ms if there is no further support for it. The content of Figure 5 seems to function as a binding assay for the soluble protein but does not directly apply to the conclusions in the main text. It should be included as a Supplementary figure.

2. Minor: In the revision, the authors have addressed several concerns about the cryo-EM model quality. For the case of the DBLMSP2 model (PDB ID 7YG2), although the overall stats have been improved, the sidechain outliers are still significant in some of the Fc μ chains (chain B, G) and the J chain (chain J). These problems should be fixed or the authors should explain in text or methods why this is not possible, e.g. quality of the map and local disorder. The location of the problems, whether inside or outside of the interface with IgM is irrelevant, as other parts of the molecule may have functional significance.

3. Minor: In their explanation for specific binding of all molecules at the Fcu1-2 position, the authors attribute it to the unique rigidity of Fcu1. However, all of the Cu4's of the core are likely to be rigid, so perhaps the answer is presently unknown.

REVIEWER COMMENTS

Reviewer #2 (Remarks to the Author):

All reviewers' comments were addressed in the updated version of the manuscript in an appropriate manner.

minor points:

-instead of using cryoSPARC please use cryoSPARC2 or 3 (whichever version was used),

We thank the the reviewer for the positive comments. Versions of cryoSPARC (v3.2), as well as other EM-related software, have been included in the revised Methods.

-please add [Å] symbol to all figure panels with local resolution,

The Å symbols have been added to the revised Supplementary Fig. 2–5.

-In supplementary Figure 3h there is a free-floating letter “f” in the lower left panel,

We thank the reviewer for carefully looking over the figures, and have made correction accordingly.

-it is sufficient to round KD, Koff and Kon results up to one decimal place (Fig.1b),

Fig. 1b has been changed as suggested.

-please expand Methods section to include more details regarding cryoEM data processing, including: number of micrographs, number of particles extracted, final number of particles in reconstruction and resolution for each map, etc.

More relevant details have been included in the revised Methods section.

-lines 235-239: consider including changes highlighted with “ ”; for example using word ‘choose’ in reference to proteins sounds a bit strange, as they are not sentient...

Fc μ 1 exhibits more rigidity when compared to Fc μ 2-5 due to its extensive interaction with the J-chain⁶, and therefore “might be more likely to bind” *P. falciparum* proteins. Furthermore, only on this side of the Fc μ -J platform the *P. falciparum* proteins can readily interact with Fc μ 2 as well. “Binding of *P. falciparum* proteins to the other side of Fc μ 1 is unlikely, as” their Fc μ 2-interacting regions would be projected towards the gap of the IgM pentamer, where J chain resides in place of an Fc μ molecule.

These changes have been included accordingly.

Reviewer #3 (Remarks to the Author):

The revised ms is improved in presentation and the overall quality of the structural models obtained from cryo-EM data is now satisfactory. The results will be of interest to the wide audience of Nature Communications. However, there is still a major issue that needs to be addressed by the authors because the mechanism for inhibiting complement protein activation is still in question.

1. Major: The authors should address the serious concern about the mechanism proposed for complement deposition and activation that was raised during the first round of review. The protein ectodomains were expressed as soluble molecules for structural study and interfere with in vitro complement activation in the assay as reported in Figure 5a, and Figure 5b illustrates a structural interpretation of the binding assay which we agree with. In vivo, however, the proteins are associated with the surface of either the merozoite or the infected blood cell membrane. Therefore, when binding IgM, they will not be available to block complement binding at the same time. The interpretation of the structures and assays cannot, without additional considerations, reflect their role in vivo. We suggested an alternative different mechanism which the authors have included, but the incorrect mechanism should be removed from the ms if there is no further support for it. The content of Figure 5 seems to function as a binding assay for the soluble protein but does not directly apply to the conclusions in the main text. It should be included as a Supplementary figure.

We thank the the reviewer for this excellent point. The original Fig. 5a has been included in Fig. 3 (as Fig. 3e) to complement the pull-down data and validate the structural analyses. The original Fig. 5b is now present as Supplementary Fig. 7c, and the relevant text has also been reframed and moved to the Discussion section (highlighted in red).

2. Minor: In the revision, the authors have addressed several concerns about the cryo-EM model quality. For the case of the DBLMSP2 model (PDB ID 7YG2), although the overall stats have been improved, the sidechain outliers are still significant in some of the Fc μ chains (chain B, G) and the J chain (chain J). These problems should be fixed or the authors should explain in text or methods why this is not possible, e.g. quality of the map and local disorder. The location of the problems, whether inside or outside of the interface with IgM is irrelevant, as other parts of the molecule may have functional significance.

The DBLMSP2 model has been further refined to improve the structural quality, as can be seen in the updated validation report.

3. Minor: In their explanation for specific binding of all molecules at the Fc μ 1-2 position, the authors attribute it to the unique rigidity of Fc μ 1. However, all of the Cu4's of the core are likely to be rigid, so perhaps the answer is presently unknown.

We thank the the reviewer again for another great point, and have expanded the relevant discussions (also highlighted in red in the revision).

REVIEWERS' COMMENTS

Reviewer #3 (Remarks to the Author):

The authors have addressed the concerns raised and the ms will be of interest to readers of Nature Communications.

REVIEWERS' COMMENTS

Reviewer #3 (Remarks to the Author):

The authors have addressed the concerns raised and the ms will be of interest to readers of Nature Communications.

We thank the reviewer for the positive comments.